# Resolving rates of mutation in the brain using single-neuron genomics

Gilad D Evrony[1,2,3,4,5†], Eunjung Lee[6,7†], Peter J Park[6,7*], Christopher A Walsh[1,2,3,4,5*]

[1]Division of Genetics and Genomics, Manton Center for Orphan Disease, Boston Children's Hospital, Boston, United States; [2]Howard Hughes Medical Institute, Boston Children's Hospital, Boston, United States; [3]Department of Neurology, Harvard Medical School, Boston, United States; [4]Department of Pediatrics, Harvard Medical School, Boston, United States; [5]Broad Institute of MIT and Harvard, Cambridge, United States; [6]Department of Biomedical Informatics, Harvard Medical School, Boston, United States; [7]Division of Genetics, Brigham and Women's Hospital, Boston, United States

**Abstract** Whether somatic mutations contribute functional diversity to brain cells is a long-standing question. Single-neuron genomics enables direct measurement of somatic mutation rates in human brain and promises to answer this question. A recent study (*Upton et al., 2015*) reported high rates of somatic LINE-1 element (L1) retrotransposition in the hippocampus and cerebral cortex that would have major implications for normal brain function, and suggested that these events preferentially impact genes important for neuronal function. We identify aspects of the single-cell sequencing approach, bioinformatic analysis, and validation methods that led to thousands of artifacts being interpreted as somatic mutation events. Our reanalysis supports a mutation frequency of approximately 0.2 events per cell, which is about fifty-fold lower than reported, confirming that L1 elements mobilize in some human neurons but indicating that L1 mosaicism is not ubiquitous. Through consideration of the challenges identified, we provide a foundation and framework for designing single-cell genomics studies.

**\*For correspondence:**
peter_park@hms.harvard.edu
(PJP); Christopher.Walsh@
childrens.harvard.edu (CAW)

†These authors contributed equally to this work

**Competing interests:** The authors declare that no competing interests exist.

## Introduction

The mechanisms that generate the immense morphological and functional diversity of neurons in the human brain have long been a subject of speculation and controversy. The immune system, with its systematic genomic rearrangements such as V(D)J recombination, and the ordered generation of random somatic mutation coupled with a selection process, have suggested appealing analogies for generating the cellular diversity of the nervous system, and have led to searches for analogous genomic diversity in the brain (*Muotri and Gage, 2006*). LINE-1 (L1) elements are endogenous retrotransposons that transcribe an RNA copy that is reverse-transcribed into a DNA copy that can then insert into a novel site in the genome, creating mutations capable of disrupting or modifying the expression of genes in which they insert or neighboring genes (*Goodier and Kazazian, 2008*). Evolutionarily, transposon mobilization is an essential cause of the generation of species diversity (*Cordaux and Batzer, 2009*), so interest in possible L1 activity during brain development was spurred by the discovery that these elements can mobilize in neuronal progenitor cells (*Coufal et al., 2009*; *Muotri et al., 2005*).

The importance of any mutation process, such as retrotransposon mobilization, in generating neuronal diversity is constrained by the rate at which mutation takes place, since if a given type of mutation occurs infrequently, it is unlikely to be a useful generator of diversity. Single-cell sequencing is a

**eLife digest** The human brain harbors perhaps the most diverse collection of cells among any organ in the body, consisting of neurons and other cells with many different shapes and behaviors. The mechanisms that create this diversity have been a long-standing area of investigation. While neurons can become more diverse through changes in the activity of genes during development and in response to experiences, it has been speculated that genetic differences among neurons may also play a role.

The complete set of genes found in an individual is known as its genome. It is often assumed that each cell in an individual's brain has an identical genome. However, mutations accumulate in cells during the lifetime of an individual such that every brain cell may in fact contain a unique set of genetic mutations. The extent and types of such genetic mutations have only recently become accessible using techniques that can examine the genomes of single-cells. Some of these genetic differences may result from the activity of short sections of DNA called retrotransposons, which can copy themselves and move to a different place in the genome. This can introduce genetic mutations that alter how the cell works.

Multiple studies have shown that retrotransposon-related mutations are present in human brain cells. Indeed, in 2015 a group of researchers suggested that every neuron in two brain regions called the cortex and the hippocampus contains as many as 16 retrotransposon-related mutations on average, which suggests that retrotransposons may play an essential role in the healthy brain. However, these findings contrasted with previous studies that had shown much fewer mutations.

Now, Evrony, Lee et al. have analyzed the data from the 2015 study that led the previous researchers to interpret some artifacts as retrotransposon mutations. Reanalysing the data confirmed that these mutations do indeed occur; however, they are around 50 times less common than had been suggested by the earlier study.

This suggests that retrotranspons are more likely to be occasional sources of rare variation or disease, rather than essential contributors to normal brain activity in humans. Further work is needed to examine the rate of these and other types of mutations in different cell types and brain regions, and at different developmental stages. However, to ensure that these studies are robust and reliable, Evrony, Lee et al. also outline a framework to aid the design and analysis of future studies.

powerful technology that has revealed and quantified previously unknown mechanisms of somatic mutation in the human brain, providing a first proof of principle for the systematic measurement of somatic mutation rates in any normal human tissue (*Evrony et al., 2012*; *McConnell et al., 2013*; *Cai et al., 2014*; *Evrony et al., 2015*; *Lodato et al., 2015*). Single-cell sequencing can therefore determine the extent to which somatic mutations diversify the genomes of cells in the brain, which is foundational to understanding their potential functional impact in normal brains and possible roles in neuropsychiatric diseases of unknown etiology (*Poduri et al., 2013*). L1 mobilization has been observed at low rates using indirect genetic techniques such as a transgenic L1 reporter in rodent brain in vivo (*Muotri et al., 2005*; *2010*) and human progenitor cells in vitro (*Coufal et al., 2009*), while studies profiling human brain bulk DNA suggested much higher rates (*Baillie et al., 2011*; *Bundo et al., 2014*; *Coufal et al., 2009*). Single-cell sequencing has been proposed as the definitive method to resolve these disparate estimates (*Erwin et al., 2014*; *Goodier, 2014*).

A recent single-cell sequencing study (*Upton et al., 2015*) reported high rates of somatic L1 retrotransposition in the hippocampus (13.7 per neuron on average) and cerebral cortex (16.3 per neuron), and suggested that L1 retrotransposition was "ubiquitous". Such a high rate of retrotransposition could present it as a possibly essential event in neurogenesis and would have major implications for brain function. Here we describe experimental artifacts that elevated the study's apparent rate of somatic retrotransposition by >50-fold. Reanalysis of their data while filtering these artifacts generates a consensus that retrotransposition does occur in developing brain but at a much lower rate consistent with prior single-cell studies (*Evrony et al., 2012*; *Evrony et al., 2015*), thereby constraining the range of possible functional roles for retrotransposition in the brain.

Our discussion of the challenges in single-cell sequencing may provide a useful framework for the design and analysis of single-cell genomics studies.

## Results

### Single-cell L1 PCR validation

Upton et al. isolated single neuronal cells from postmortem human brains and amplified their genomes using the MALBAC method (*Zong et al., 2012*). They then profiled human-specific L1 elements (L1Hs) using their RC-seq method (*Shukla et al., 2013*; *Upton et al., 2015*) that captures and amplifies both the 5' and 3' ends of L1Hs elements via oligonucleotide hybridization and PCR, hence providing sequence data that identify L1Hs insertion sites in the genome.

Although whole genome amplification methods have remarkable abilities to amplify picogram quantities of DNA from a single cell into microgram quantities, chimeric DNA molecules falsely linking unrelated DNA fragments are well known to arise during single-cell genome amplification (*Macaulay and Voet, 2014*). Chimera artifacts also arise during ligation and PCR steps of sequencing library preparation (*Kircher et al., 2012*; *Quail et al., 2008*), processes integral to the RC-seq method. Chimeric sequences can create a DNA fragment connecting a LINE element to an unrelated portion of the genome, creating the appearance of biological LINE mobilization (*Figure 1—figure supplement 1A–B*). Sequence analysis of Upton et al.'s putative PCR-validated candidates (Table S2 of Upton et al.) demonstrates that more than half of them (7 of 13) are chimera artifacts that could not have been generated by the process of L1 mobilization (*Supplementary file 1*). Some chimeras originated immediately downstream of germline L1Hs/L1Pa elements that are incapable of retrotransposition, because the L1 elements are truncated, are from an old, inactive L1 subfamily, or contain numerous inactivating mutations (*Figure 1B*; *Supplementary file 1*). In other cases, L1 5' and 3' junction chimeras originating from distinct L1 elements were misinterpreted as two ends of the same L1 (*Figures 1B–C*; *Supplementary file 1*). Some candidates lack poly-A tails (*Figure 1C*; *Supplementary file 1*), a key feature of retrotransposon insertions. Although the remaining 6/13 PCR-validated insertions lack clear evidence of being chimeras, the possibility cannot be excluded based on the limited PCR validation performed, and they are likely also chimeras because each is supported by only 1 or 2 sequencing reads (see below). The presence of chimeric artifacts among a set of insertions passing limited PCR validation supports the importance of additional careful analysis of candidate L1 sequences to help define more accurate rates of retrotransposition.

Upton et al. performed 3' junction PCR validation for 10 RC-seq candidates for which they had detected the 5'end, and while 3' junction validation is technically straightforward, it failed for all 10. Whereas the authors attribute their validation failure to poly-A tails obstructing PCR amplification, poly-A tails do not obstruct PCR: the single-cell RC-seq method used by Upton et al. itself entails three PCR steps in which L1 3' junctions are amplified, and 3' junction PCR is the standard validation approach used in L1 studies (*Ewing and Kazazian, 2010*; *Grandi et al., 2012*; *Huang et al., 2010*; *Iskow et al., 2010*). The failure of all 10/10 3' junction validation attempts suggests a high prevalence of false-positives among RC-seq candidates detected with only 5' junction reads, which represent 27% of all candidate insertions.

### Definitive validation of somatic insertions

"Full-length" validation is the most accurate method to screen out false-positive candidate somatic insertions. A bona fide L1 insertion creates two genome breakpoints at the insertion site, one on each end of the insertion (5' and 3'), while a chimera has only one breakpoint at the called insertion site (*Figure 1A*). Full-length cloning validation, in which the entire L1 insertion is amplified in a single DNA molecule spanning both breakpoints (using primers based in the genomic sequence flanking the L1), is therefore the only way to confirm that both breakpoints are present in the same DNA molecule as a bona fide insertion, as opposed to two different chimeric molecules (*Figure 1A*; *Figure 1—figure supplement 1*). However, Upton et al. did not perform full-length cloning validation on any insertion. Two independent chimeras can even occur by chance in two different DNA molecules/copies whose non-L1 sequences overlap the same genomic locus, giving the false appearance of a target site duplication (TSD) (*Figures 1B–C*; *Supplementary file 1*). In single-cell sequencing,

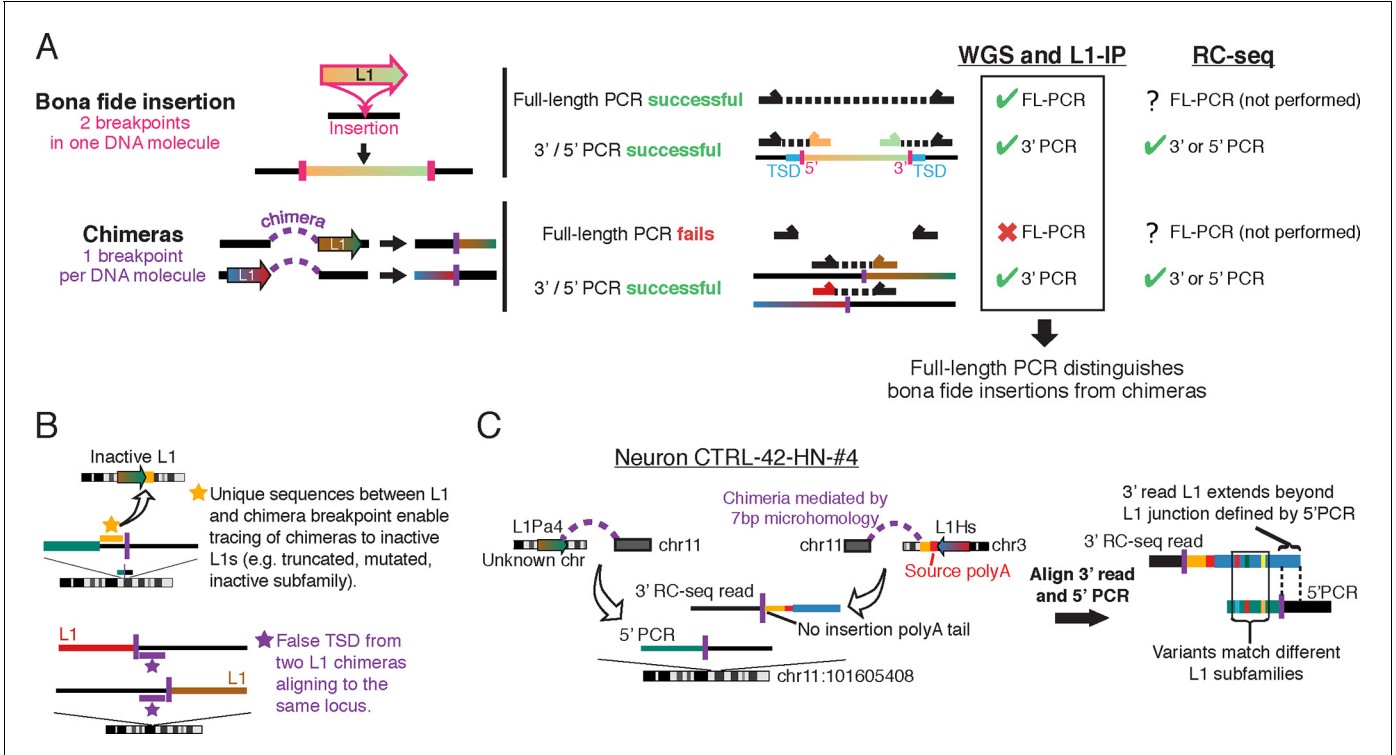

**Figure 1.** Chimera artifacts in RC-seq. (**A**) Full-length (FL) PCR using primers flanking the insertion site is necessary for definitive validation of somatic insertions in single cells in the setting of chimeras. One breakpoint per chimera DNA molecule refers to the breakpoint of the candidate insertion being analyzed since a DNA molecule can in principle have multiple different chimera events each involving different loci (which would be unlikely to create a structure that would validate by FL-PCR). For most RC-seq candidates, Upton et al. did not attempt 3' or 5' PCR for the computationally identified junction and only performed this for the opposite junction. (**B**) Top schematic illustrates one of several methods for identifying L1 chimeras in next-generation sequencing data such as RC-seq. Bottom schematic illustrates how two independent chimeras aligning to the same locus appear to have a TSD. (**C**) An example somatic insertion candidate that passed Upton et al. single-junction PCR validation but derived from two independent chimera artifacts. Yellow region is non-L1 sequence from chromosome 3 that allows tracing of the chimera to its source. L1Pa4 is an inactive L1 subfamily (*Hancks and Kazazian, 2012*). See *Supplementary file 1* ('RC-seq | Somatic L1 PCR' sheet) for analyses of all somatic candidates passing Upton et al. PCR validation. See also *Figure 1—figure supplement 1*.

The following figure supplement is available for figure 1:

**Figure supplement 1.** Overview of single-cell L1 profiling methods and chimeras in the context of genome amplification, analysis, and PCR validation

especially when read count is not used to filter candidates (see below), full-length cloning of at least some candidate insertions is important to exclude chimera artifacts.

Instead of full-length cloning validation (*Evrony et al., 2015*; *Stewart et al., 2011*), Upton et al. carried out multiple 5' junction PCR reactions per candidate using primers spaced every 500 bp along the ~6000 bp L1. Multiple PCR reactions, each with an L1 primer matching hundreds to thousands of genomic loci, introduces additional mechanisms for generating false-positives. Some candidates required nested PCR with 62 PCR cycles, an extremely high level of amplification, suggesting that the targets are chimeras present at very low level in the single-cell amplified DNA.

The MALBAC method employed by Upton et al. for single-cell genome amplification probably precludes definitive full-length cloning validation of some insertions, suggesting it is not an ideal method for studying retrotransposition. MALBAC produces short amplicons (0.5–1.5kb) compared to multiple displacement amplification (MDA) (10–50 kb amplicons) (*Dean et al., 2002*; *Zhang et al., 2015*; *Zong et al., 2012*), so insertions longer than ~1.5 kb (~15–30% of somatic L1 insertions in human cancer studies [*Helman et al., 2014*; *Lee et al., 2012*; *Tubio et al., 2014*]) would not be efficiently validated in MALBAC single cells (*Figure 1—figure supplements 1A and C*).

The many chimeras among 'validated' somatic insertions come about because Upton et al. relied solely on computational analysis of contig sequences (assembled from short sequencing reads) to filter chimeras and labeled most remaining candidates as putative somatic insertions. However, sequence analysis of short contigs that do not span the full length of insertions can identify chimeras but cannot rule them out. For example, 5' junction chimeras originating from inside an L1 element or 3' junction chimeras originating from within the poly-A tail cannot be distinguished from true insertion breakpoints by sequence analysis alone and require further experiments (i.e., full-length cloning) (*Figure 1A*). Furthermore, even some chimeras that can be identified computationally were not filtered: one recurrent type of chimera artifact accounts for 16% of their candidates detected only at a 5' junction (*Supplementary file 1*; see example candidate chr11:112602973). Further analyses, as well as long-read sequencing (e.g., PacBio), may reveal additional ways to remove chimeras computationally by sequence analysis alone; but with short-read sequencing, even ideal sequence-based filtering algorithms cannot filter chimeras originating from within L1.

## L1 and chimera read count distributions

A core principle of next-generation sequencing analysis is the use of read counts to distinguish true mutations from artifacts that inevitably arise during DNA sequencing (*Robasky et al., 2014*; *Sims et al., 2014*). Multiple reads supporting a mutation serves the same role as replication does in any scientific experiment, increasing the confidence that the finding is not an artifact. This is especially important in single-cell sequencing where chimeric DNA artifacts are more prevalent than in standard sequencing (*Macaulay and Voet, 2014*). Essentially all major mutation-detection algorithms use the signal strength (read count and often other parameters) of known true mutations and false-positive events to predict the likelihood that individual candidate mutations are real and to determine a signal cutoff (*Chen et al., 2009*; *Cibulskis et al., 2013*; *DePristo et al., 2011*; *Mills et al., 2011*; *1000 Genomes Project Consortium, 2012*). However, Upton et al. did not employ a read count filter or signal model and therefore considered candidate insertions supported by only a single read as equivalent to the smaller number of candidates with higher read support. As a result, 97% (4634/4759) of their single-cell insertion calls were supported by a single Illumina sequencing read and 99.6% by 1 or 2 reads; 94% of their >320,000 candidates from 'bulk' DNA were also supported by only 1 read (*Figure 2A*).

Upton et al.'s rationale for not using read counts in their analysis is their suggestion that in their single-cell RC-seq method, chimeras appear at higher read counts than true insertions such that nearly all true insertions would be detected by only 1 read. This proposal can be tested using the read count distribution of a 'gold standard' mutation set. In single-cell samples, somatic insertions should appear at the same signal level distribution as germline known non-reference L1 insertions (KNR), which are population-polymorphic L1 insertions absent from the reference human genome but identified in prior L1 studies. Germline KNR insertions share the same sequence characteristics as somatic insertions (*Helman et al., 2014*; *Lee et al., 2012*; *Tubio et al., 2014*) and bear no distinguishing feature that would lead to different read counts. Therefore, KNR insertions can be used to directly test Upton et al.'s model that true insertions preferentially appear at lower read counts than chimeras.

Using RC-seq single-cell germline KNR insertion data provided by the authors upon request, we found that KNR insertions were detected by much higher read counts than candidate somatic insertions. In single-cell RC-seq samples, 53%, 24% and 20% of the 4049 calls of high-confidence gold-standard KNR insertions were detected with $\geq 3$, $\geq 20$ and $\geq 40$ reads per sample, respectively; only 32% were detected with only 1 read (*Figure 2A*; *Figure 2—figure supplement 1A*). In contrast, 97% (4634/4759) of single-cell somatic insertion candidates were detected with only 1 read and only 0.4% (20/4759) with $\geq 3$ reads (*Figure 2A*). The strikingly higher read depths of gold-standard germline KNR L1 insertions relative to somatic insertion candidates in the same experiment is consistent with the vast majority of claimed somatic insertions not corresponding to bona fide insertions.

Analysis of RC-seq L1 junction detection rates provides additional evidence that nearly all somatic candidates are false-positives (*Figure 2B*). 11% of single-cell KNR insertion calls were detected at both L1 (5' and 3') junctions, whereas >250-fold less— only 0.04% (2/4682)— of single-cell somatic insertion candidates were detected at both junctions. Sequence analysis shows 8 of the 12 hippocampal single-cell somatic candidates detected at both junctions (including candidates in which each junction was detected in a different sample) are chimera artifacts (*Supplementary file 1*). The

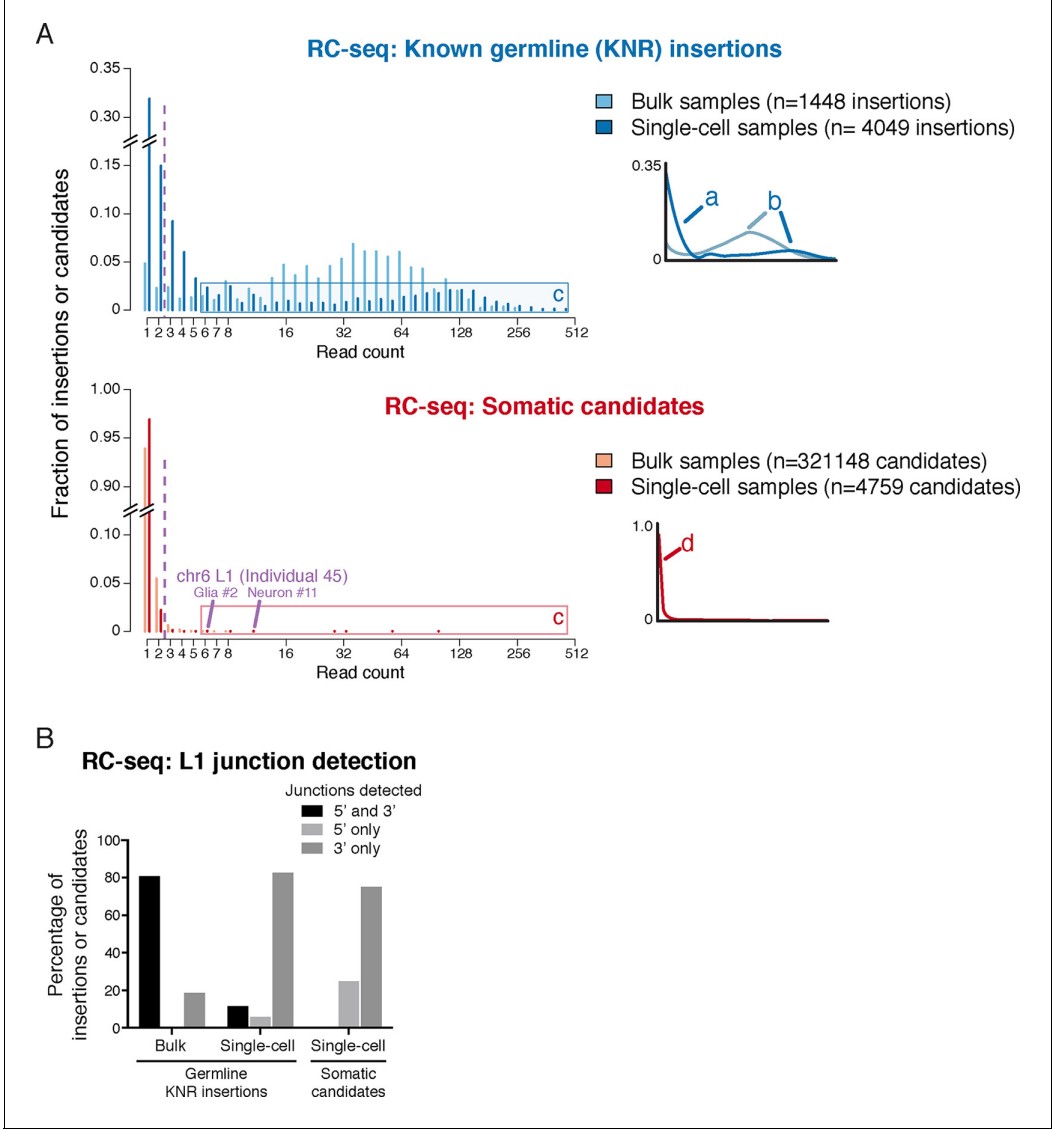

**Figure 2.** RC-seq read count distributions and junction detection rates of somatic insertion candidates are inconsistent with known true germline insertions. (**A**) RC-seq read count distributions of bulk and single-cell germline known non-reference (KNR) insertion and somatic candidate calls (see 'Materials and methods'). Inset schematics and labels 'a' to 'd' illustrate key findings. Label 'a' (inset schematic) points to the subset of KNR insertions appearing at low read counts in single-cells, distinct from the distribution of KNR insertions in bulk samples, due to dropout/non-uniformity at length scales < 30 kb inherent to MALBAC amplification (Appendix 2). These factors are also responsible for the broader distribution of higher read count KNR insertions (label 'b') in single-cell versus bulk samples. Areas labeled 'c' in the top and bottom graphs highlight the population of single-cell KNR insertions at high read counts that is absent from single-cell somatic candidates. KNR insertions present in a single copy per cell (chrX insertions in male samples) show the same pattern (*Figure 2—figure supplement 1A*). Instead, single-cell somatic candidates appear at very low read counts (label 'd', inset schematic). The likely bona fide insertion detected in two single cells on chromosome 6 is labeled and appears at high read count relative to other somatic candidates. Purple dashed line indicates threshold of > 2 reads used for calculation of somatic retrotransposition rates. See also *Figure 2— figure supplement 1*. (**B**) L1 junction detection rates in bulk and single-cell RC-seq (see 'Materials and methods'). Fewer KNR insertions are detected at both (5' and 3') junctions in single-cell versus bulk samples due to MALBAC amplification dropout/non-uniformity. A significantly lower fraction of single-cell somatic candidates are detected at both junctions relative to single-cell KNR insertions, confirming the vast majority of somatic candidates are false-positives.

The following figure supplement is available for figure 2:

**Figure supplement 1.** Read count distributions of known germline insertions in different L1 profiling methods.

remainder cannot be excluded as chimeras without full-length PCR validation. Furthermore, RC-seq bulk somatic candidates have a non-canonical distribution of large TSD sizes, inconsistent with nearly all prior L1 research (Appendix 1). Analysis of 10 randomly selected candidates with large (>50 bp) TSDs found all were chimera artifacts (*Supplementary file 1*).

## Corrected RC-seq somatic insertion rates

A more plausible RC-seq somatic insertion rate can be calculated using a read count threshold calibrated to germline high-confidence KNR insertions as a gold standard. A read count threshold of > 2 optimizes sensitivity and specificity, maintaining detection of 53% of true positive KNR insertion calls across all single cells (*Figure 2A*) and a per-cell KNR detection sensitivity of 24% (*Figure 3A*), while excluding ~99.6% of false-positive calls (see 'Materials and methods'). Only 20 somatic insertion candidates supported by >2 reads were detected across all 170 cells, and 12 of these were chimeras upon further sequence analysis (*Supplementary file 1*). The remaining 8 candidates yield a sensitivity-corrected somatic insertion rate estimate of 0.19 per cell, with no significant difference in rates between cell types (hippocampal neurons and glia, cortical neurons, and AGS hippocampal neurons) (p = 0.98, ANOVA) (*Figure 3B*). 95% of single cells did not have any somatic insertion candidates (excluding chimeras) supported by >2 sequencing reads. These RC-seq somatic insertion rates are quite consistent with rates previously estimated by L1 insertion profiling (L1-IP) in single cortical and caudate neurons (0.07 ± 0.15 (SD); p = 0.54, ANOVA) (*Evrony et al., 2012*), and using single-neuron whole-genome sequencing (0.18 ± 0.47 (SD); p = 0.37, ANOVA) (*Evrony et al., 2015*), suggesting a notable consensus by three methods confirming that somatic L1 insertions are present in human brain, but fewer than one per average genome.

Notably, 2 of the 8 somatic insertion candidates detected following read count filtering correspond to a single, likely bona fide L1Hs insertion in neuron #11 (12 reads) and glial cell #2 (6 reads) from the hippocampus of individual 45 (*Supplementary file 1*). This intergenic insertion shows RC-seq reads capturing both 5' and 3' junctions bearing all the hallmarks of a true retrotransposition event: a TSD, poly-A tail, and a 3' transduction that traces its source to a population-polymorphic (KNR) L1 on chromosome 2 that was identified in a prior L1 profiling study (*Iskow et al., 2010*). This same somatic insertion was also detected in glial cells #7 and #8 of the individual, each with 2 reads. Upton et al. highlighted this insertion for its detection in multiple cells but did not note its high-signal level—this candidate had the 5[th] and 9[th] highest read counts of all 4759 somatic candidate calls (*Figure 2A*). This clonal retrotransposon event also showed >1 read in all 4 cells in which it was detected and was detected at both 5' and 3' junctions. The basic signal characteristics of this one clear somatic insertion event make it dramatically different from those of the thousands of other somatic insertions proposed by Upton et al. (*Figures 2A–B*).

## Single-cell MALBAC performance

MALBAC-amplified single cells profiled by RC-seq had reduced performance relative to bulk RC-seq in terms of gold-standard KNR insertion read counts and junction detection rates (*Figures 2A–B*), and had significantly lower sensitivity for KNR insertions (higher dropout) than L1 profiling of MDA-amplified single cells (*Figure 3A*; *Figure 3—figure supplement 1*). We therefore further studied the quality of Upton et al.'s single cells and the performance of the MALBAC method (*Zong et al., 2012*) that the authors used for single-cell genome amplification.

Analysis of Upton et al.'s pre-RC-seq whole-genome sequencing of MALBAC-amplified single cells shows that at genomic scales < 50 kb (high-resolution view), which includes the size range of retrotransposons and single-nucleotide variants (SNV), there are systematic ~1 kb peaks of high genome amplification separated by troughs of low amplification or complete dropout (*Figure 4A*). These peaks and troughs often occur in the same locations as in MALBAC single cells from an unrelated study by *Zong et al. (2012)* (*Figure 4A*), suggesting that this non-uniformity in genome amplification is inherent to MALBAC. In contrast, MDA single cells show significantly better uniformity of genome amplification at these size scales (*Figure 4A*). The non-uniformity of MALBAC at genomic scales encompassing the size range of retrotransposon elements likely explains the subset of true KNR insertions appearing at low read counts (*Figure 2A*) and the low sensitivity (high allelic dropout) of single-cell RC-seq (*Figure 3A*). It also explains MALBAC's lower overall breadth of genome-wide

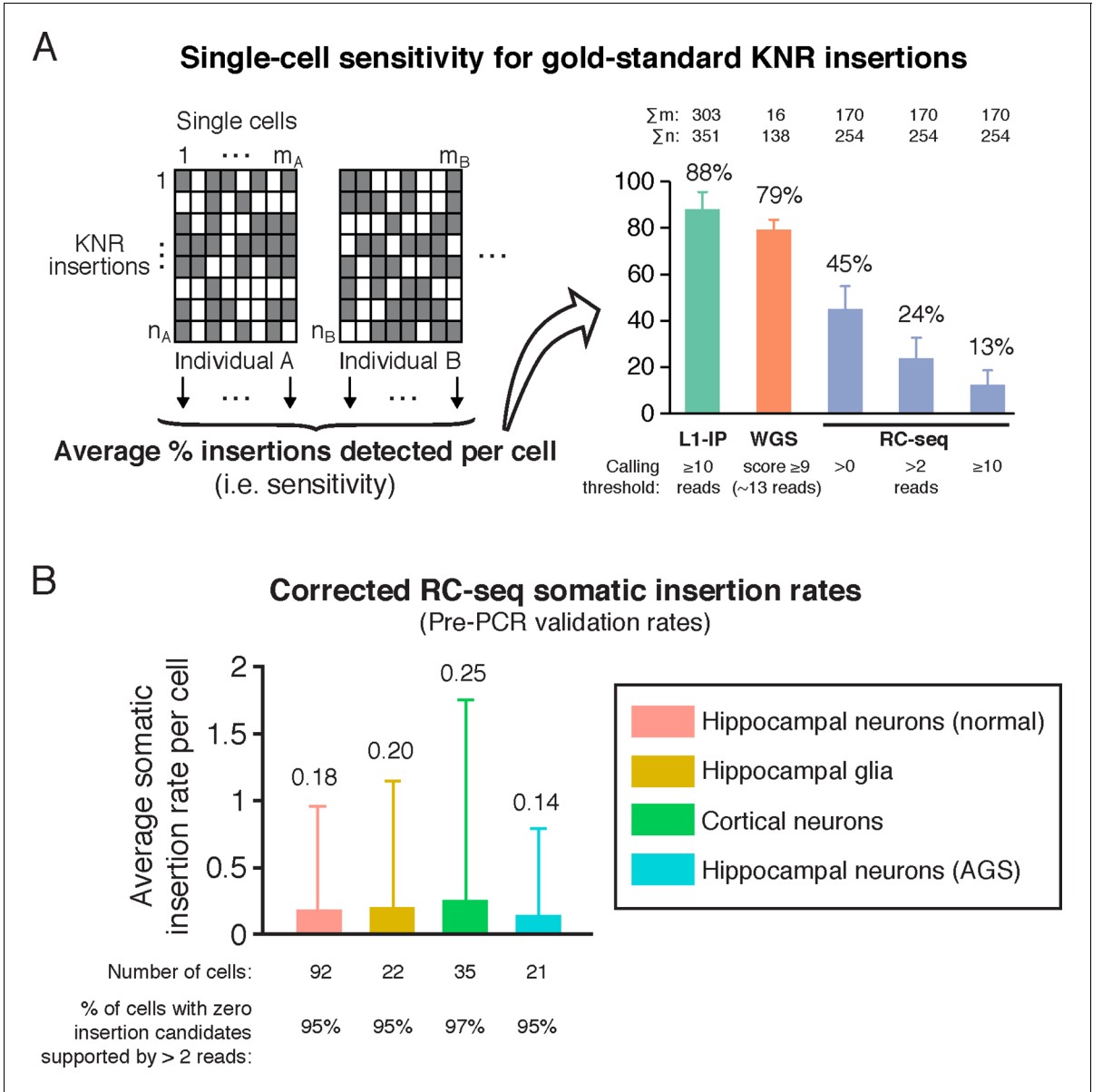

**Figure 3.** RC-seq sensitivity for gold-standard true insertions and corrected RC-seq somatic insertion rates. (**A**) Average sensitivity of single-cell RC-seq for gold-standard KNR insertions at different read count thresholds. Sensitivity of single-cell L1-IP (*Evrony et al., 2012*) and single-cell WGS (*Evrony et al., 2015*) are shown for comparison. Note, the average number of uniquely mapped reads in the targeted enrichment methods of L1-IP and RC-seq are 3.2 and 16.7 million reads, respectively, so L1-IP achieves higher sensitivity than RC-seq with fewer reads even with a more liberal read count threshold for RC-seq. Gold-standard KNR insertions are defined for each single-cell method as in *Figures 2A* and *Figure 2—figure supplement 1B–C*. Error bars ± SD. As illustrated in the schematic on the left, $\Sigma m$ is the number of single cells in the study (i.e. $m_A+m_B+...$), and $\Sigma n$ is the number of gold-standard KNR insertions used to calculate sensitivity across the profiled individuals (i.e. $n_A+n_B+...$; as seen in the schematic, $\Sigma n$ increases as more individuals are profiled). See also *Figure 3—figure supplement 1*. (**B**) Average RC-seq somatic insertion rates per cell. These are pre-PCR validation rates, since Upton et al. did not attempt PCR validation for these somatic candidates. The percentage of cells without any candidates (above the threshold of >2 reads and after excluding chimeras) is shown. See *Supplementary file 1* ("RC-seq | Somatic L1 >2 reads" sheet) for analysis of all somatic candidate sequences. Error bars ± SD.

The following figure supplement is available for figure 3:

**Figure supplement 1.** Single-cell sensitivity of L1-profilng methods for gold-standard germline KNR insertions.

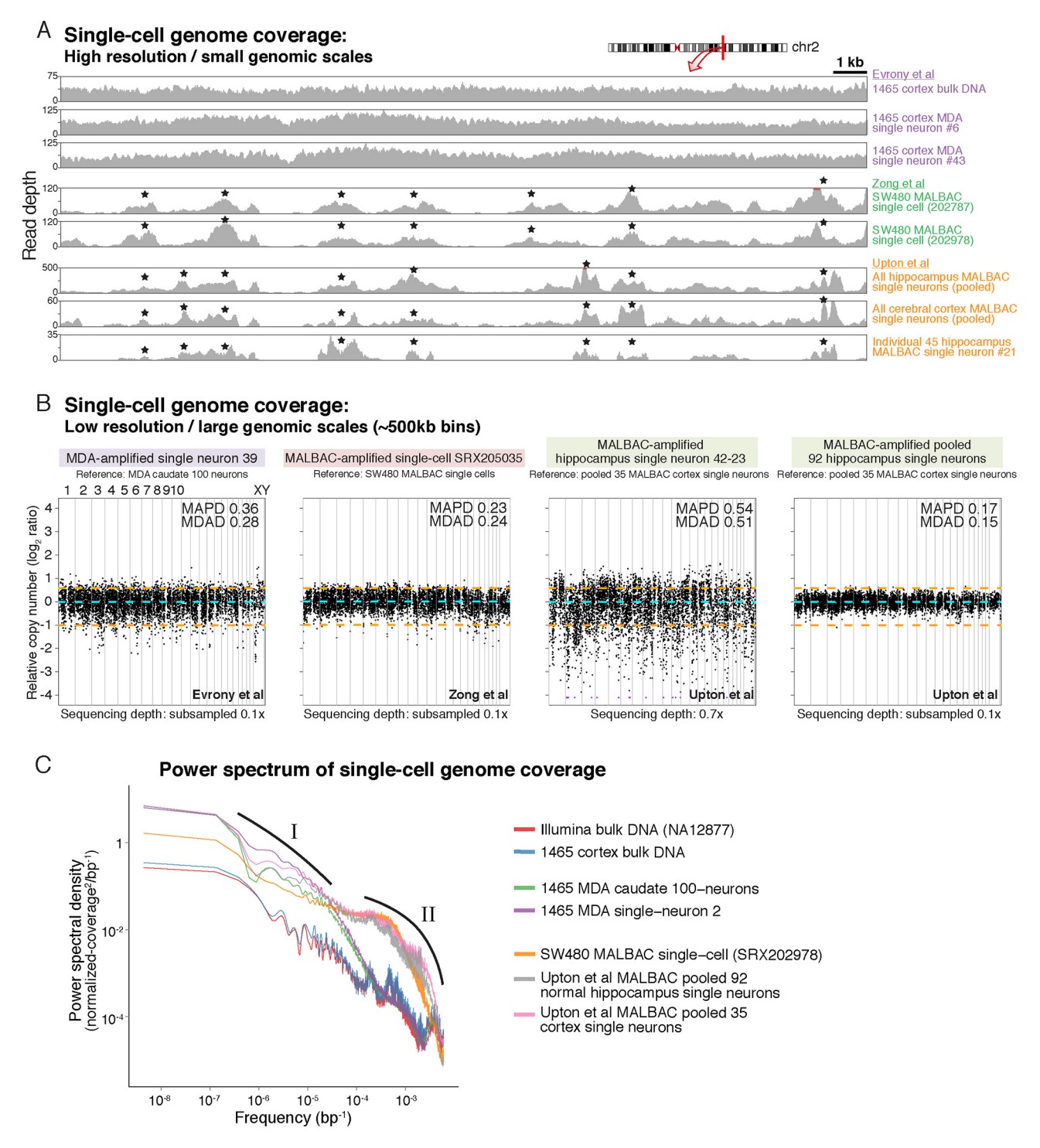

**Figure 4.** MDA and MALBAC single-cell genome amplification uniformity. (**A**) High-resolution coverage plots of MDA single neurons (*Evrony et al., 2015*) and MALBAC single cells from *Zong et al. (2012)* and *Upton et al. (2015)*. MALBAC samples show significant non-uniformity with systematic high peaks (stars) and troughs of genome amplification. MALBAC single neurons from Upton et al. were pooled from hippocampus (n = 92 cells) and cortex (n = 35 cells) of normal individuals to produce high-coverage samples for the plots. Pooling eliminates stochastic noise of individual cells but preserves systematic non-uniformity inherent to MALBAC. Area shown is chr2:155,815,550–155,848,725 encompassing the region of one of the single-cell RC-seq somatic L1 candidates detected with both 5' and 3' junctions (chr2:155,823,436). Red lines mark off-scale peaks. (**B**) Low-resolution (~500 kb bin) genome-wide coverage plots of representative single cells from the above studies. MALBAC single cells from Zong et al. have significantly better uniformity at these scales than MDA single neurons as measured by median absolute pairwise deviation (MAPD) and median absolute deviation from

*Figure 4 continued on next page*

*Figure 4 continued*

the median (MDAD) scores (lower scores indicate higher uniformity) (*Cai et al., 2014*; *Evrony et al., 2015*). In contrast, individual MALBAC single cells from Upton et al. have significantly lower quality than both MALBAC single cells from Zong et al. and MDA single neurons. Pooling of all 92 normal hippocampus single neurons from Upton et al. achieves high uniformity (low MAPD/MDAD scores), indicating the low quality of individual single cells from Upton et al. is due to stochastic noise, likely from factors preceding MALBAC amplification. Note, high-coverage MALBAC and MDA samples from Zong et al. and Evrony et al. were subsampled to a lower read depth similar to read depth of Upton et al. samples, confirming prior analyses showing uniformity quality metrics are not affected by sequencing depth in low resolution analyses (*Evrony et al., 2015*). (C) Power spectral density (y-axis), which reflects the degree of read depth variability (uniformity) as a function of genomic spatial frequency (x-axis). Higher spatial frequencies (right side of x-axis) reflect smaller genomic scales (i.e. higher resolution, as in *Figure 4A*), and lower spatial frequencies (left side of x-axis) reflect larger genomic scales (i.e. lower resolution, as in *Figure 4B*). Plots show differences in MDA and MALBAC genome amplification uniformity across genomic scales: MDA single cells have greater read depth variability at larger genomic scales than MALBAC single cells [label I], while MALBAC has greater read depth variability at smaller genomic scales < 30 kb [label II] (i.e. scale of SNVs, small indels, retrotransposons; frequencies > $\sim 3.5 \cdot 10^{-5}$ bp), consistent with high-resolution coverage plots (*Figure 4A*). MALBAC single cells from Upton et al. were pooled to obtain high-coverage samples for the analysis. Plots for individual 1465 and SW480 samples were calculated in *Evrony et al. (2015)* and are presented again for comparison to Upton et al. samples. Additional unrelated bulk sample NA12877 is plotted for comparison. See Appendix 2 for additional details, *Figure 4—figure supplement 1* for average MAPD/MDAD scores of single cells and additional coverage plots, and *Figure 4—figure supplement 2* for basic genome coverage statistics.

The following figure supplements are available for figure 4:

**Figure supplement 1.** MDA and MALBAC single-cell quality and low-resolution genome-wide amplification uniformity.
**Figure supplement 2.** MDA and MALBAC genome coverage.

coverage at nucleotide resolution (i.e. higher locus dropout) relative to MDA (*Figure 4—figure supplement 2A*).

At larger genomic scales of ~500 kb bins (low-resolution view), MALBAC single cells from Upton et al. show significantly lower quality and higher variability among individual cells than both MALBAC-amplified single cells from Zong et al. and MDA-amplified single neurons (*Figure 4B*; *Figure 4—figure supplement 1A–B*). Pooling of all 92 normal hippocampus single neurons from Upton et al. shows performance commensurate with MALBAC single cells from Zong et al. (*Figure 4B*; *Figure 4—figure supplement 1A–B*), indicating that the low quality of Upton et al.'s single cells may be due to stochastic factors preceding MALBAC, such as poor tissue quality, rather than MALBAC itself. Of note, at these genomic scales, MALBAC single cells from Zong et al. have high reproducibility and better uniformity of genome coverage than MDA (*Figure 4B*; *Figure 4—figure supplement 1A–B*) (*Evrony et al., 2015*), enabling MALBAC's better performance in detection of large copy number variants (*Hou et al., 2013*). Power spectral density measuring amplification uniformity across all genomic scales confirmed better uniformity of MALBAC at large genomic scales (> 30 kb) and better uniformity of MDA at small genomic scales (< 30 kb) (*Figure 4C*) (*Evrony et al., 2015*; *Zhang et al., 2015*). The above and additional analyses are discussed further in Appendix 2. Altogether, these results: a) suggest MALBAC and low quality single cells as contributors to single-cell RC-seq sensitivity loss; b) emphasize the importance of single-cell quality control at genomic scales relevant to the studied mutation type; and c) confirm that MALBAC and MDA each have advantages at different genomic scales and for different mutation types but that MALBAC is not especially well-suited for retrotransposon studies.

## Discussion

Here, we have shown that L1 mosaicism is not "ubiquitous" in the hippocampus and that somatic insertion rates in the recent paper by Upton et al. were elevated > 50-fold due to the informatic analysis and a lack of definitive validation.

### Read counts of true insertions versus chimeras

To justify not using a read count filter, Upton et al. state that "in single-cell RC-seq libraries, putative chimeras are disproportionately likely to amplify efficiently and accrue high read depth" (*Upton et al., 2015*). In other words, they are suggesting that their method preferentially amplifies noise (chimeric sequences) instead of signal (true insertions). We could find no precedent

or chemical explanation for why PCR or next-generation sequencing would preferentially amplify chimeras, since there are no sequence features distinguishing chimeras from true insertions in small DNA fragments that would cause preferential overamplification of the former in single-cell RC-seq. In fact, prior single-neuron sequencing studies and chimera rates of Illumina libraries and MALBAC show directly that chimeras are not preferentially amplified relative to true genomic sequence fragments and true insertions (Appendix 3; *Figure 2—figure supplement 1B–C*; *Figure 3—figure supplement 1*).

Indeed, the use of read counts for mutation analysis is integral to one of the prime purposes of single-cell sequencing, a technology whose development was motivated by two goals: (a) tracking which somatic mutations are present together in the same cells to enable lineage tracing; and (b) achieving higher signal to noise ratios for somatic mutations, i.e. true mutation to false-positive read count ratios. In single-cell sequencing, somatic mutations appear on average at the same signal level as germline heterozygous mutations (i.e. 50% of reads at the locus), while the fraction of false variant reads at a locus (e.g. sequencing errors, library PCR mutations, chimeras) is the same on average regardless of the number of cells sequenced. Accordingly, decreasing the number of cells pooled for sequencing increases the signal to noise ratio of somatic mutations (see *Figure 5* for a simplified mathematical framework for single-cell sequencing). Therefore, calling mutations supported by only a single sequencing read is counter to a key feature and objective of single-cell sequencing. Furthermore, although Upton et al. present qPCR experiments as additional evidence for their findings, it is important that the originators of that qPCR method consider single-cell analysis as definitive (*Erwin et al., 2014*), and qPCR results are affected by target L1 specificity (Appendix 4).

Finally, we emphasize that the bioinformatic and validation approach led to the inflated somatic insertion rate, but not the RC-seq L1 hybridization capture method itself. Our analysis suggests that RC-seq capture, if used with an appropriate single-cell amplification method, careful signal modeling based on true insertions, and rigorous PCR validation, would likely enable cost-effective, high-throughput retrotransposon profiling comparing favorably with other methods such as L1-IP.

## Somatic retrotransposition rates in the brain

The corrected RC-seq retrotransposition rate is significant as it aligns to a wholly different regime of potential functional roles for retrotransposition in the brain (rare normal variation and rare disease) rather than a "ubiquitous" role. This corrected rate is consistent with rates measured in vitro in neuronal progenitors (*Coufal et al., 2009*) and is consistent with the absence of significant somatic L1 insertions in brain tumors (*Helman et al., 2014*; *Iskow et al., 2010*; *Lee et al., 2012*). These rates do not rule out that there may be rare individuals in whom a somatic L1 insertion affects a gene in enough cells to cause a sub-clinical or overt phenotype, or that elevated L1 rates may occur in particular individuals or disease states. Future single-neuron genomic studies will resolve the rates and mosaicism frequencies of all classes of somatic mutation across the diversity of cell types, regions, and developmental timepoints in the brain.

Single-cell genomic analysis has enabled the first systematic measurement of somatic mutation rates in the body but entails additional challenges spanning molecular biology to bioinformatics. Our findings suggest the following elements may aid future single-cell genomics studies: a) choosing a single-cell amplification method suitable for the studied mutation type; b) objective metrics evaluating genome amplification coverage, uniformity, dropout, and chimera rates at spatial scales and genomic elements relevant to the mutation type; c) use of gold-standard germline mutations and chimera rates to build a signal model for calling mutations; and d) stringent validation experiments. Retrotransposons offer unique advantages as a starting point for developing single-cell genomics methods due to their characteristic sequence signatures allowing definitive validation even when present in only one cell. The lessons learned from the study of somatic retrotransposition are therefore broadly applicable for the nascent field of single-cell genomics.

# Materials and methods

## Data sources

Sequencing data of single-cell whole-genome sequencing (WGS) experiments from Upton et al. were obtained from the European Nucleotide Archive with accession PRJEB5239. Single-cell RC-seq

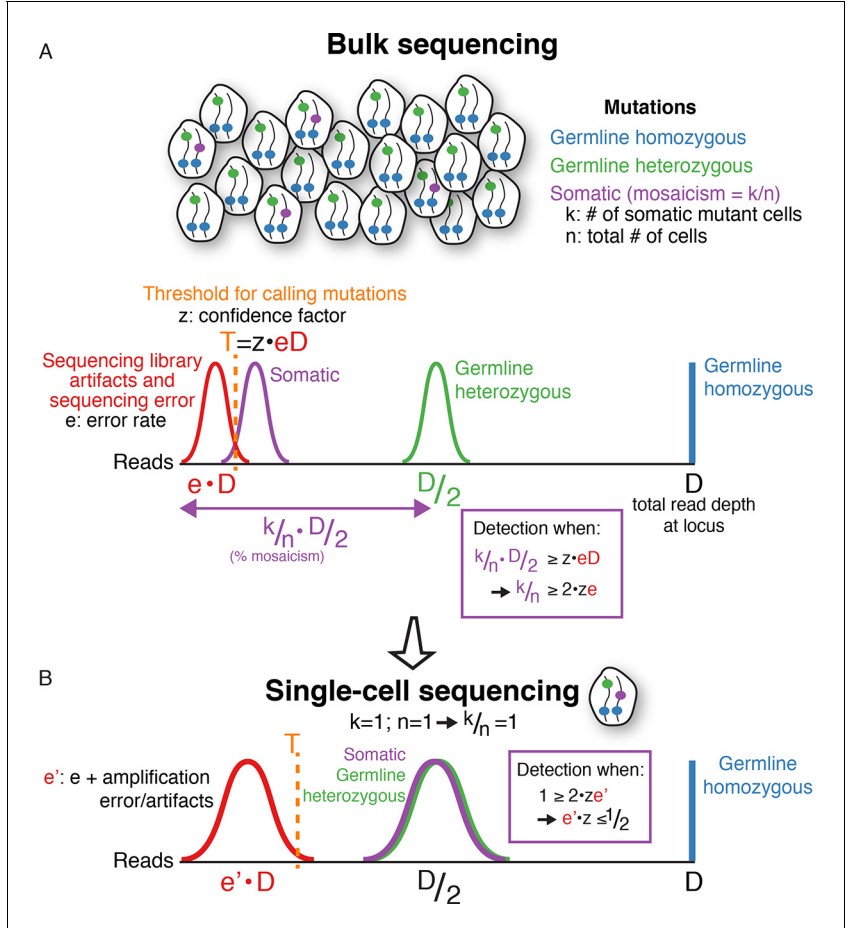

**Figure 5.** A mathematical framework for single-cell sequencing. (**A**) In bulk sequencing, a somatic mutation present in *k* out of *n* cells pooled together for sequencing (i.e. mosaicism of *k/n*), with read coverage *D* at the mutation locus, will be detected on average in *k/n·D/2* reads with a variance depending on sampling error; i.e. the number of reads detecting the mutation correlates linearly with the percent mosaicism. In contrast, germline heterozygous and homozygous variants are present in *D/2* and *D* reads, respectively. Due to sequencing artifacts and sequencing errors, a mutation must be detected above a threshold number of reads, *T*, which also depends on the sequencing depth, *D*, since errors occur at rates, *e*, that are a constant fraction on average of the total number of reads (*T=z·e·D*; *z* is a constant chosen based on desired detection sensitivity and specificity). The fraction of error reads, *e*, is a constant on average that is independent of total sequencing depth, *D*, because library artifacts and sequencing errors occur at rates that are independent of total sequencing depth. The threshold, *T*, can be reduced with methods reducing sequencing error, but errors are still present in any current sequencing technology. Combining equations simplifies to k/n ≥ 2·z·e. This means that the mosaicism of a somatic mutation must be at least twice the sequencing error rate (or more, depending on the confidence factor) for detection to be possible in bulk DNA sequencing, regardless of sequencing depth. Below a certain level of mosaicism that depends on the sequencing error rate, detection is unlikely. Note: for simplicity, the height of the histograms (# of mutations) is scaled to the same height, and the equations do not include variance terms. (**B**) In single-cell sequencing, somatic mutations are present at the same signal level on average as germline heterozygous variants (i.e. D/2, since k/n = 1), enabling detection of low mosaicism mutations that would otherwise be below detection thresholds of bulk sequencing due to sequencing error. Due to whole genome amplification, single-cell sequencing also leads to greater variance in mutation and error signal level distributions (non-uniform amplification and dropout) and entails additional artifacts not present in bulk sequencing, which increases the noise level, *e'*, but still a lower level on average than true heterozygous mutations. However, the signal distribution of artifacts may still overlap that of true mutations, necessitating careful bioinformatics and modeling of error and true mutation signals along with rigorous validation. Note, for simplicity, the equations here do not include variance terms and bioinformatic modeling usually includes additional parameters other than read count illustrated here. Single-cell sequencing does not achieve increased sensitivity for somatic mutations without cost, because to detect a given mutation with *k/n* mosaicism, more than *n/k* single cells may need to be sequenced. The benefit of single-cell sequencing is not to reduce sequencing costs, but rather its ability to overcome limitations due to sequencing error rates on the minimum mosaicism detectable and maintaining information as to which somatic mutations are found within the same cell, which enables lineage tracing.

somatic candidate data (including sequences) and bulk RC-seq KNR insertion data were obtained from Upton et al. supplemental tables (2015). Single-cell RC-seq KNR (germline polymorphic) insertion data (read counts and junction detection rates) and bulk RC-seq somatic L1 candidate data were provided by Geoffrey Faulkner upon request.

Sequencing data of single cells from *Evrony et al. (2015)* and *Zong et al. (2012)* used in MDA and MALBAC performance analyses were obtained from those studies as described in *Evrony et al. (2015)*.

High-coverage bulk DNA sequencing of individual N12877 shown in power spectral analysis (*Figure 4C*) was obtained from the NCBI Sequence Read Archive with accession ERX069504.

## RC-seq candidate sequence analysis

RC-seq insertion candidate sequences were analyzed with the aid of standard tools, including the UCSC genome browser (*Kent et al., 2002*), Blat (*Kent, 2002*), NCBI BLAST (http://blast.ncbi.nlm.nih.gov/Blast.cgi), RepeatMasker (*Smit et al., 2010*), RepBase (*Jurka et al., 2005*), ClustalW2 (*Larkin et al., 2007*), and L1Xplorer (*Penzkofer et al., 2005*).

## Whole-genome sequencing read alignment

MALBAC and Illumina sequencing adaptors were trimmed from sequencing reads of Upton et al. MALBAC single-cell WGS samples using the 'phacro' tookit (http://sourceforge.net/projects/phacro) (*Hou et al., 2013*) with default settings and the MALBAC adaptor: GTGAGTGATGGTTGAGGTC TTGTGGAG. The phacro toolkit was created by the team that developed MALBAC specifically for trimming MALBAC adaptors from MALBAC samples, including the 8bp degenerate 'N' sequence following the adaptor.

After adaptor trimming, Upton et al. WGS data was aligned to the hs37d5 human genome reference (1000 Genomes Project human genome reference based on the GRCh37 primary assembly) with bwa (*Li and Durbin, 2009*) as in *Evrony et al. (2015)*. PCR duplicates were removed as in *Evrony et al. (2015)*.

## Whole-genome sequencing coverage and performance analyses

High resolution genome coverage plots (*Figure 4A*), low resolution (~500 kb bin) genome coverage plots (*Figure 4B* and *Figure 4—figure supplement 1*), power spectral density analysis (*Figure 4C*), subsampling genome coverage analysis (*Figure 4—figure supplement 2A*), and Lorenz curves (*Figure 4—figure supplement 2A*) were calculated and plotted as in *Evrony et al. (2015)* (results summarized in Appendix 2). Plots for samples from individual 1465 and SW480 MALBAC samples in power spectral density analysis (*Figure 4C*), subsampling analyses (*Figure 4—figure supplement 2A*), and Lorenz curves (*Figure 4—figure supplement 2B*) were already calculated in *Evrony et al. (2015)* and are presented again in this paper to allow comparison to Upton et al. single-cell samples. Median absolute pairwise deviation (MAPD) and median absolute deviation from the median (MDAD) scores of single-cell quality were calculated in ~500 kb equal-read bins as in *Evrony et al. (2015)*.

High-resolution genome coverage plots (*Figure 4A*), power spectral density analysis (*Figure 4C*), subsampling genome coverage analysis (*Figure 4—figure supplement 2A*), and Lorenz curves (*Figure 4—figure supplement 2B*) were calculated after pooling all single neurons from normal individual hippocampi (n = 92 cells) to create a high-coverage dataset (48x), since the WGS sequencing depth of individual cells in Upton et al. are not sufficient for high-resolution analyses. A high-coverage (5x) pooled sample of all single neurons from normal individual cerebral cortex (n = 35 cells) was also created for the high-resolution genome coverage plot (*Figure 4A*) and power spectral density analysis (*Figure 4C*). Low-resolution genome coverage plots and analyses (*Figures 4B* and *Figure 4—figure supplement 1*) were performed for individual hippocampus single neurons and also separately for the pooled hippocampus single-neuron sample. Low-resolution genome coverage plots of Upton et al. single cells used the pooled cerebral cortex single-neuron sample as a copy number reference. Note that pooling to achieve higher coverage datasets would only improve genome coverage statistics since as samples are pooled, stochastic noise present in individual cells cancels out, leaving systematic noise due to MALBAC and providing a view of MALBAC amplification performance.

Low-resolution genome coverage plots and analyses of MDA single neurons (*Evrony et al., 2015*), SW480 MALBAC single cells (*Zong et al., 2012*), and the pooled hippocampus MALBAC single-neuron sample (*Upton et al., 2015*) were also calculated after subsampling these high coverage samples to lower read depths (*Figures 4B* and *Figure 4—figure supplement 1*), confirming as in *Evrony et al. (2015)* that low-resolution genome coverage plots and statistics are minimally affected by increasing read depth > 0.1x. Therefore the results and conclusions of low-resolution genome coverage analyses are not due to lower sequencing depth for Upton et al. single cells relative to MALBAC and MDA samples from other studies, as the conclusions are the same after subsampling MALBAC and MDA samples from other studies to lower read depth than Upton et al. single cells.

Chromosome X bins in low-resolution genome coverage plots of single cells from individual CTRL-36 (female) and the pooled hippocampus single-neuron sample (which includes CTRL-36 female neurons) (*Figure 4—figure supplement 1A*) were corrected in each sample to the median of all bins in chromosome X of the sample, since the pooled cortex single neurons used as a copy number reference derived from male samples so chromosome X bins would have inflated copy number without correction. Chromosome Y bins of each CTRL-36 (female) hippocampus single neuron were set to a $\log_2$ relative copy number of 0 so that they do not affect genome coverage statistics, since CTRL-36 female neurons do not have a Y chromosome and complete dropout of Y-chromosome bins would skew (i.e. make worse) genome coverage statistics. Chromosome Y bins of the pooled hippocampus single-neuron sample were also normalized to the median of chromosome Y bins in the sample, since this pooled sample includes CTRL-36 female neurons that do not have a Y chromosome.

Discordant and clipped read statistics for Upton et al. single-cell WGS samples (Appendix 3) were calculated as in *Evrony et al. (2015)*. Discordant and clipped read statistics for MALBAC single-cell samples from *Zong et al. (2012)* and MDA single-cell samples from *Evrony et al. (2015)* were already calculated in *Evrony et al. (2015)*.

## Read count histograms

Read count histograms of somatic insertion candidates and germline known non-reference (KNR) insertions (*Figure 2A* and *Figure 2—figure supplement 1*), which are insertions detected in prior L1 profiling studies that are absent from the human genome reference, were constructed as described below for each L1 profiling method. Upton et al. acknowledge the importance of KNR gold-standard insertions by using them to estimate the sensitivity of their method, but they did not present the distribution of KNR insertion read counts in single cells, which is essential data for calling somatic insertion candidates and evaluating candidate veracity.

Read count histograms plot the *per sample* read counts of candidates and insertions, not their total read count across all samples, which controls for the number of samples profiled per individual and for candidates/insertions present in multiple samples (necessary for comparing germline KNR insertions that are present in many samples to somatic candidates).

*RC-seq KNR read count histograms (Figure 2A and Figure 2—figure supplement 1A):* Single-cell RC-seq KNR read counts were obtained from data provided by Geoffrey Faulkner upon request. Bulk RC-seq KNR read counts were obtained from the 'Polymorphic L1' sheet of Table S2 in *Upton et al. (2015)*. The gold-standard set of germline KNR insertions plotted for single cells in *Figure 2A* and *Figure 2—figure supplement 1A* consists of insertions identified in prior non RC-seq L1 profiling studies (i.e. insertions with a prior study annotated in the 'Database?' column of Upton et al. tables) that were detected with $\geq 40$ reads in both bulk samples of the individual (considering detection only in bulk samples corresponding to the individual from whom the single cell derived). Insertions that were detected only in a prior RC-seq study ("Published RC-seq?' column) but not in a prior non RC-seq study (empty 'Database?' column) were not included in *Figure 2A* and *Figure 2—figure supplement 1A* since it is preferable to define a gold standard set of true mutations detected by independent methods. Nevertheless, read count histograms that also include KNR insertions that were identified only in prior RC-seq studies produced nearly identical histograms (data not shown). Therefore, whether or not KNR insertions found only in prior RC-seq studies are included has negligible effect. Bulk KNR insertion read count histograms in *Figure 2A* and *Figure 2—figure supplement 1A* show KNR insertions detected at any read count (i.e. $\geq 1$ read), since there is no independent gold-standard reference as to which KNR insertions are present in bulk samples of the profiled individuals, and using a $\geq 40$ read cutoff would mask the underlying read count distribution by showing

only insertions appearing at high read counts. In any case, the key comparison for evaluating RC-seq somatic candidate veracity is between single-cell KNR insertions and single-cell somatic candidates, not between single-cell KNR insertions and bulk KNR insertions. The latter comparison is useful for assessing the quality of single cells versus bulk samples and the effect of MALBAC amplification.

Note that germline KNR insertion dropouts in single cells (read counts of 0 for germline KNR insertions in single cells of an individual known to have the KNR insertion based on bulk samples) are not included in the read count histograms since single-cell dropout rates affect both KNR insertions and somatic insertions. While for KNR insertions the true state (presence/absence) in each cell is known, the true state is unknown for somatic insertions. Therefore, in order to compare germline KNR insertion and somatic candidate read count distributions, KNR dropout calls must be excluded.

Also, note that the read count distribution of gold-standard KNR insertions in single-cell RC-seq is bimodal (*Figure 2A*), with a population of high read count calls and a population of low read count calls. Although KNR insertions appear at lower read depth in single cell RC-seq relative to bulk RC-seq samples and show a bimodal distribution with ~1/3 of calls detected by only one read (*Figure 2A*), this does not affect the conclusion that the vast majority of single-cell RC-seq somatic insertion candidates are false-positives: only 20 of the 4759 somatic candidates were detected with > 2 reads across all 170 single cells and half of true somatic insertions are expected to be detected at this level based on KNR insertion read counts. However, it does predict that ~1/3 of true somatic insertions would be detected with 1 read. This bimodal distribution of KNR read counts in single-cell RC-seq is due to, both: a) high variability (non-uniformity) in single-cell MALBAC genome amplification at the length scale of L1 insertions (data not shown; and see *Evrony et al. (2015)*: Note S1, 'Coverage variability analyses' section, Figure S6, and Figure S7, as well as *Zhang et al. (2015)* for details of non-uniformity at small length scales < 30 kb inherent to MALBAC); and b) allelic dropout stemming from low-quality of Upton et al. single neurons. The MAPD (median absolute pairwise deviation) metric reflects uniformity of genome coverage at large genomic scales (~500 kb bins), with lower MAPD scores indicating better uniformity. Upton et al. single neurons have mean MAPD scores of $0.53 \pm 0.16$ (SD), compared to MAPD $0.18 \pm 0.06$ for MALBAC-amplified single cells from *Zong et al. (2012)* and MAPD $0.33 \pm 0.06$ for MDA-amplified single neurons from *Evrony et al. (2015)*.

Furthermore, the lower overall read counts of KNR insertions in single cells relative to bulk samples is also partly due to ~3fold lower total reads per sample on average for single-cell samples versus bulk samples. This highlights a further issue when read count filters are not used, in that there is no normalization for different total reads per sample.

Somatic insertions are present in a single copy in the genome (i.e. heterozygous or hemizygous) in cells harboring the mutation. Most germline KNR insertions (~75%) are present in a single copy per cell as well, since most are in the heterozygous state in individuals of the population. This supports the use of KNR insertions as a reference for the expected read count distribution (and signal distribution of other parameters) of somatic insertions. The evidence that most KNR insertions that are present in an individual are heterozygous is based on measured allele frequencies and genotypes of KNR insertions in prior population studies of L1 polymorphism: a) In the 1000 Genomes project studying mobile element polymorphism (*Stewart et al., 2011*), genotyping of a large number of L1 KNR insertions (see Table S4 in that study) found an average heterozygosity of 0.85 in individuals harboring the insertions (i.e. number of individuals heterozygous/(number heterozygous + number homozygous) for each KNR insertion, averaged across all KNR insertions). The average allele frequency of these insertions was 0.26; b) In *Iskow et al. (2010)*, the average allele frequencies of KNR insertions found by dideoxy sequencing was 0.22 (table S1 in that study) and < 0.2 for insertions found by 454 sequencing (*Figure 2F* in that study), corresponding to a heterozygosity rate for KNR insertions of at least 0.88 in individuals harboring each insertion (i.e., allele frequency p = 0.22; heterozygosity in individuals with the insertion = $2pq/(p^2+2pq)$) assuming insertions are in Hardy-Weinberg equilibrium. Prior studies have shown L1 insertion genotypes are almost always consistent with Hardy-Weinberg equilibrium (*Badge et al., 2003*; *Myers et al., 2002*; *Seleme et al., 2006*); c) Ewing and Kazazian (2011) also analyzed the 1000 genomes data and found a KNR insertion allele frequency < 0.2 (*Figure 1B* in that study), corresponding to an average heterozygosity >0.89 for KNR insertions present in an individual; d) *Huang et al. (2010)* estimate an allele frequency of chromosome X KNR insertions of 0.58 and an allele frequency of 0.38 for a set of KNR insertions identified by whole-genome profiling, corresponding to an average heterozygosity of 0.59 and 0.77,

respectively in individuals harboring the insertions; d) 75% (105/140) of the KNR insertions detected in individual 1465 in *Evrony et al. (2015)* (gold-standard KNR insertions detected in both bulk samples of the individual) are heterozygous or hemizygous (*Evrony et al., 2015*); e) in dbRIP (*Wang et al., 2006*), the average heterozygosity of polymorphic insertions among individuals with the insertion is 0.46 (with an average allele frequency of 0.59). This shows that most KNR insertions in an individual are heterozygous and present in a single copy per cell. We also plotted the RC-seq read count histograms of a pure set of single-copy KNR insertions– those found on chromosome X in male samples– and found a similar distribution of read counts as the full KNR set, with most insertions still detected by multiple reads in single cells: 65%, 12%, and 11% were detected with $\geq$3, $\geq$20 and $\geq$40 reads per sample (*Figure 2—figure supplement 1A*).

*RC-seq somatic candidate read count histogram (Figure 2A):* Single-cell RC-seq somatic candidate read counts were obtained from the 'Somatic L1' sheet of Table S2 in *Upton et al. (2015)*. Bulk RC-seq somatic candidate read counts were provided by Geoffrey Faulkner upon request.

*WGS KNR read count histogram (Figure 2—figure supplement 1B):* The gold-standard KNR insertion set for the WGS read count histogram is defined as insertions detected in both bulk samples (cortex and heart) of the individual with the following parameters (see *Evrony et al. (2015)* for details of parameters): a) $\geq$ 2 RAM reads on each side of the breakpoint; b) $\geq$ 4 clipped reads supporting the insertion call; c) estimated target-site duplication or deletion $\leq$ 50 bp in size in the absence of a poly-A tail, or $\leq$ 250 bp in size if a poly-A tail was detected; d) at least half of clipped reads at the insertion site aligned to $\pm$ 2 bp of the insertion breakpoint; e) the insertion was detected in prior independent L1 profiling studies from other groups (see *Evrony et al. (2015)* for list of prior L1 profiling studies used).

*L1-IP KNR read count histograms (Figure 2—figure supplement 1C):* The gold-standard KNR insertion set for the L1-IP read count histograms was defined as insertions detected with a confidence score $\geq$ 0.5 in at least half of the bulk samples of the individual and detected in prior independent L1 profiling studies of other groups (see *Evrony et al. (2012)* for list of prior L1 profiling studies used).

## RC-seq L1 junction detection rates

The percentage of RC-seq insertions and candidates detected at only the 5', only the 3', or both 5' and 3' L1 junctions (*Figure 2B*) were obtained as follows:

Germline KNR junction detection data for bulk and single-cell RC-seq samples were provided by Geoffrey Faulkner; these data annotated for each individual sample and each KNR insertion which L1 junctions were detected (5', 3', or both). Junction detection rates of both bulk and single-cell germline KNR insertions shown in *Figure 2B* are for the same high-confidence KNR insertion set defined for the single-cell KNR read count histogram in *Figure 2A* (see 'Read count histograms' in the prior section of the 'Materials and methods'). The numerator and denominator units of bulk and single-cell RC-seq KNR junction detection rates are KNR insertion *calls*, not KNR insertions; i.e. for a hypothetical KNR insertion detected in samples A, B, and C, each of these 3 *calls* is counted separately because the detection of a KNR insertion in each sample is independent of other samples.

Single-cell RC-seq somatic candidate junction detection data were obtained from the 'Somatic L1' sheet of Table S2 in *Upton et al. (2015)*. 5'-only detected candidates are those with a negative alignment in the 'Sense L1' column but no antisense read or a negative alignment in the 'Antisense L1' column but no sense read. 3'-only detected candidates are those with a positive alignment in the 'Sense L1' column but no antisense read or a positive alignment in the 'Antisense L1' column but no sense read. Candidates detected at both 5' and 3' junctions are those with both sense and antisense reads. Note that the 'single-cell somatic candidate' junction data available in Table S2 of Upton et al. annotates junction detection per candidate (regardless of the number of cells in which the candidate was detected), in contrast to the 'single-cell KNR insertion' junction data that annotates junction detection for each individual sample in which the insertion was detected. Since 'single-cell somatic candidate' junction detection data is only available annotated per candidate rather than per cell, somatic candidates detected in multiple cells may skew the true junction detection rates and preclude comparison to 'single-cell KNR insertion' rates. Therefore, to allow comparison between 'single-cell somatic candidate' and 'single-cell KNR insertion' junction detection rates, the 'single-cell somatic candidate' junction detection rates in *Figure 2B* are for candidates detected in only one cell and excludes those detected in multiple cells. Nevertheless, even when including candidates found

in more than one cell (i.e. considering both junctions as detected even when each was detected in a different single cell), only 0.4% (21/4728) of single-cell somatic candidates were detected at both junctions– still >25-fold less than the rate for single-cell KNR insertions (11%) and similar to the single-cell somatic candidate rate of 0.04% (2/4682) calculated when excluding candidates found in multiple cells.

## RC-seq somatic retrotransposon insertion rate calculation

Briefly, somatic insertion rates were calculated by first counting the number of somatic candidates detected with > 2 reads. Sequences of candidates were then manually examined and definite chimeras were excluded (*Supplementary file 1*). In each cell, the remaining number of candidates was adjusted for that cell's sensitivity for gold-standard KNR insertions. Insertion rates per cell type (*Figure 3B*) are an average of the rates across all single cells of that type. Below is a full explanation of the insertion rate calculations:

RC-seq somatic retrotransposon insertion rates were calculated using RC-seq read counts of the gold-standard germline KNR insertion set to guide read count filtering. A read count threshold was chosen that would optimize the number of true (germline KNR and somatic) insertions above the threshold (sensitivity) while minimizing the number of false-positive calls (specificity). Sensitivity for true insertions at any given read count threshold was estimated *per single cell* using the single-cell RC-seq KNR insertion read count data provided by Geoffrey Faulkner. Sensitivity was calculated as the fraction of high-confidence germline KNR insertions present in the individual (i.e. insertions detected with $\geq$ 40 reads in both bulk samples of the individual, and identified in prior non RC-seq L1 profiling studies with a prior study annotated in the 'Database?' column of Upton et al. tables), that were detected in the single cell above the read count threshold. Specificity at any given read count was estimated using the read count distribution of all single-cell somatic candidate calls ('Somatic L1' sheet of Table S2 in *Upton et al. (2015)*) since nearly all are false-positives. As discussed in the main text and in the following paragraph, the latter assumption is valid because of the discrepancy between the read count distributions of KNR insertions versus somatic candidates.

As discussed above in the 'Read count histograms' section, the single-cell RC-seq KNR insertion read count distribution is bimodal due to non-uniformity of MALBAC amplification, with high and low read-count sub-populations (*Figure 2A*). Finite mixture modeling can estimate the proportion of the read count distribution that belongs to each sub-population. Finite mixture modeling estimates the high and low read count sub-populations comprise 1/3 and 2/3 of the single-cell KNR insertion distribution, respectively. In contrast, the read count distribution of single-cell somatic candidates is unimodal, concentrated at low signal with nearly all (99.6%) candidates having $\leq$ 2 reads (*Figure 2A*). Intuitively, the absence of a high-signal component in the somatic candidate read count distribution indicates nearly all somatic candidates are false-positives. Therefore, the somatic candidate read count distribution can be treated essentially as a false-positive distribution for the purposes of deciding on an optimal read count threshold. More formally, any single-cell somatic candidate distribution is a mixture of two subpopulations: false-positive candidates (e.g. chimeras) and true somatic insertions. A finite mixture model can estimate the proportion of somatic candidates that derives from a true somatic insertion subpopulation, using a model of the high read-count component of the true-positive KNR insertion distribution as a guide. This analysis estimates a negligible fraction (< 0.5%) of single-cell somatic candidates are true somatic insertions. Consequently, we can consider the read count distributions of KNR insertions and somatic candidates as reflecting true and false-positives, respectively. This then allows calculation of estimated sensitivity loss and specificity gain at increasing read count thresholds.

Increasing the read count threshold from >0 to >1 read reduces the per-cell sensitivity for true (KNR) insertion calls from an average of 45% to 31% (a 32% reduction) while reducing the estimated number of false-positive calls by ~97%. Further increasing the threshold to > 2 reads reduces the sensitivity for true insertion calls to 24% (a further 23% reduction) and reduces false-positive calls by an estimated additional ~84% relative to the > 1 read threshold– still a large improvement in specificity with a relatively modest reduction in sensitivity. Increasing the read count threshold further to > 3 reads leads to diminishing returns in terms of improved specificity– 18% reduction in sensitivity with 35% reduction in false-positive calls relative to the > 2 read threshold– reflecting the fact that nearly all somatic candidate (mostly false-positive) calls are at read counts of 1 and 2. Therefore a read count threshold of > 2 reads was chosen, which maintains detection of 53% of KNR insertion

calls across all single cells and a per-cell KNR detection sensitivity of 24%, while excluding an estimated 99.6% of false-positive calls.

Once the > 2 reads count threshold was chosen, for each single cell the number of somatic insertion candidates detected with > 2 reads was counted. Candidate sequences were then manually examined and candidates that were definite chimeras were excluded (see 'RC-seq | Somatic L1 > 2 reads' sheet in *Supplementary file 1* for sequence analyses of all candidates). For each cell, the remaining number of somatic candidates in the cell was then corrected for the sensitivity for gold-standard KNR insertions achieved in that same cell, i.e. dividing the number of somatic candidates by the fraction of KNR insertions detected in the cell above the chosen threshold, using the gold-standard KNR insertion reference of the individual from whom the single-cell derived, as described above. This final number was the estimated pre-PCR validation somatic insertion rate for the cell, since Upton et al. did not attempt PCR validation for these somatic candidates. Insertion rates per cell type (*Figure 3B*) are an average of the rates across all single cells of that type.

Further justification for the > 2 reads threshold is shown by estimates of the pre-PCR validation somatic insertion rate at read thresholds of >1, > 3, and > 4 reads. At a read threshold of >1 read, the estimated pre-PCR validation rate across all cells is 2.4 ± 3.3 (SD) per cell prior to manual examination of candidates for chimeras. Adjusting for the chimera rate of 12/20 seen at the > 2 read threshold (since the chimera rate at a >1 read threshold could only be greater), gives a rate of 1.1 ± 1.4 insertions per cell. At read thresholds of > 3 and > 4 reads, the estimated pre-PCR validation rates across all cells are 0.38 ± 1.35 and 0.44 ± 1.69 (SD), respectively, per cell prior to manual examination of candidates for chimeras. Adjusting for chimera rates of 8/13 and 7/12, respectively, seen in manual examination of the candidates (*Supplementary file 1*) yields pre-PCR validation rate estimates of 0.15 ± 0.52 and 0.18 ± 0.70, respectively, for the > 3 and > 4 read thresholds. These are similar to the estimate of 0.19 ± 0.97 at a > 2 read threshold. In summary, the pre-PCR validation rates across all single cells at >1, > 2, > 3, and > 4 read thresholds (after excluding chimeras) are 1.1, 0.19, 0.15, and 0.18, respectively. This shows stability of the rate estimate at thresholds of > 2 reads or more, a result of the fact that the vast majority of chimeras appear with 1 or 2 reads, while most true insertions appear at higher read counts. The stability of the rate estimate above thresholds of > 2 reads supports the use of the > 2 read threshold, which optimizes sensitivity and specificity. In contrast, the somatic rate calculated at the > 1 read threshold is higher than the rates calculated at > 2, > 3, and > 4 reads and a significant overestimate of the true rate for two reasons: a) the > 1 read threshold begins to overlap the false-positive chimera distribution, so most candidates at the > 1 read threshold are chimeras. This is confirmed by the read count histogram analyses of KNR insertions and somatic candidates discussed above– namely that there is no discernible population of high read count candidates in the read count distribution of somatic candidates as there is in the KNR insertion read count distribution (*Figure 2A*), so the population of somatic candidates at read counts of 1 and 2 are nearly all false-positives; b) this is a pre-PCR validation rate. The somatic insertion rate estimate obtained at a > 1 read threshold is therefore an overestimate that would be confirmed as such after proper PCR validation, while the rate obtained at a > 2 read threshold is a more accurate pre-PCR validation rate estimate.

The somatic L1 retrotransposition rate for single neurons from *Evrony et al. (2015)* was calculated for comparison to the RC-seq rate. 16 single neurons were sequenced in *Evrony et al. (2015)*, but the rate was estimated from the 14 single neurons that were selected randomly for sequencing. The 2 remaining cells in which L1 #1 was detected (neurons 2 and 77) (*Evrony et al., 2015*) were excluded from the rate estimate, because they were a priori chosen for whole-genome sequencing as positive controls known to harbor somatic L1 insertions previously detected by the L1-IP method in *Evrony et al. (2012)*. Therefore, the calculated rate reflects the 2 of the 14 single neurons (neurons 6 and 18) that harbor the same L1 #2 clonal insertion (*Evrony et al., 2015*). The L1 somatic insertion rate estimate of each neuron was corrected for the neuron's sensitivity for KNR insertions.

## L1-IP 3' PCR validation

3' junction PCR validation of 48 L1-IP candidates with low read counts (*Supplementary file 1*, sheet 'L1-IP | low-read-count') was performed as described in *Evrony et al. (2012)*. The L1-IP computational pipeline was rerun on raw data from *Evrony et al. (2012)* after removing any read count filter. 24 candidates were randomly selected from all candidates detected by only 1 read, and another 24 candidates were randomly selected from all candidates.

## Acknowledgements

Single-cell KNR insertion data and bulk somatic L1 candidate data were provided by Geoffrey Faulkner. We thank David Symer for providing supplemental data from prior work on TSD sizes in HeLa cells. GDE is supported by NIH MSTP grant T32GM007753 and the Louis Lange III Scholarship in Translational Research. EL is supported in part by the Eleanor and Miles Shore Fellowship and the William Randolph Hearst Fund. CAW is supported by the Manton Center for Orphan Disease Research and grants from the NINDS (R01 NS079277 and R01 NS032457). CAW is a Distinguished Investigator of the Paul G Allen Family Foundation, and an Investigator of the Howard Hughes Medical Institute.

## Additional information

### Funding

| Funder | Grant reference number | Author |
| --- | --- | --- |
| National Institutes of Health | | Gilad D Evrony<br>Eunjung Lee<br>Peter J Park<br>Christopher A Walsh |
| National Institute of Neurological Disorders and Stroke | R01 NS079277 and R01 NS032457 | Gilad D Evrony<br>Christopher A Walsh |
| Howard Hughes Medical Institute | | Gilad D Evrony<br>Christopher A Walsh |
| Boston Children's Hospital | | Gilad D Evrony<br>Christopher A Walsh |
| Louis Lange III Scholarship in Translational Research | | Gilad D Evrony |
| Eleanor and Miles Shore Fellowship | | Eunjung Lee |
| William Randolph Hearst Fund | | Eunjung Lee |
| Paul G. Allen Family Foundation | | Christopher A Walsh |
| National Institutes of Health | T32GM007753 | Gilad D Evrony |

The funders had no role in study design, data collection and interpretation, or the decision to submit the work for publication.

### Author contributions

GDE, EL, Conception and design, Acquisition of data, Analysis and interpretation of data, Drafting or revising the article; PJP, CAW, Conception and design, Drafting or revising the article

## Additional files

### Supplementary files

• Supplementary file 1. Sequence analysis of RC-seq somatic retrotransposon insertion candidates, and validation results of low read-count L1-IP candidates. Detailed sequence analyses of RC-seq candidates can be found within the embedded word documents in the 'Sequence Analysis' column of each RC-seq spreadsheet (double-click to open documents). "RC-seq | Somatic L1 PCR" sheet presents sequence analyses of all RC-seq candidates passing Upton et al.'s RC-seq PCR validation; see candidate chr6:37821198 analysis document for example schematic. "RC-seq | Somatic L1 > 2 reads" sheet presents sequence analyses of all RC-seq somatic L1 candidates detected by > 2 reads in a single cell; this is the set of somatic candidates above the read count threshold chosen for calculation of corrected RC-seq somatic insertion rates. This sheet contains sequence analysis of the likely bona fide insertion at chr6:58481778. "RC-seq| Somatic hippoc. 5+3 jxn" sheet presents sequence analyses of all RC-seq somatic L1 candidates detected in hippocampal single neurons at both 5' and

3' junctions. "RC-seq| Bulk somatic L1 TSD>=50" sheet presents sequence analyses of 10 randomly selected RC-seq somatic L1 candidates detected in bulk samples with a TSD of at least 50 bp; see candidate chrX:85583069 analysis document for example schematic. In RC-seq sheets, columns with new analyses have blue column headers. Remaining columns with white headers (candidate meta-data and sequences) were obtained as follows: candidate metadata and sequences for "RC-seq | Somatic L1 PCR", "RC-seq | Somatic L1 > 2 reads", and "RC-seq| Somatic hippoc. 5+3 jxn" sheets were obtained from Table S2 ("Somatic L1" and "Somatic L1 PCR" sheets) of Upton et al.; candidate metadata and sequences for the "RC-seq| Bulk somatic L1 TSD>=50" sheet were obtained from the full RC-seq bulk somatic insertion table provided by Geoffrey Faulkner.

"L1-IP | low-read-count" sheet presents candidate information and validation results of 24 randomly selected L1-IP candidates detected by only 1 read and 24 L1-IP candidates randomly selected without any read count filter. Candidates were obtained from L1-IP data from *Evrony et al. (2012)*. All candidates failed PCR validation, illustrating true insertions do not preferentially appear at low read counts in L1-IP and the importance of using read counts to filter candidates.

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

## Appendix 1: Non-physiologic TSD sizes of RC-seq bulk somatic insertion candidates

The distribution of TSD sizes of RC-seq bulk hippocampus somatic candidates contains many large TSDs, a finding inconsistent with nearly all prior L1 research. Of the RC-seq bulk somatic candidates detected at both 5' and 3' junctions, 44%, 37%, and 30% of candidates were detected with TSDs $\geq$ 30, $\geq$ 40, and $\geq$ 50 bp in size, respectively. In contrast, prior L1 research has shown TSD sizes < 30 bp for nearly all insertions. The authors point to a prior study of in vitro retrotransposition in HeLa cells that found large TSDs ($\geq$ 50 bp in 20% of insertions) (*Gilbert, et al., 2005*) and hypothesize the same occurred in neuronal progenitors due to chromatin properties shared with HeLa cells, such as "pervasive euchromatinization". Large TSDs greater than 50 bp have only been found at an appreciable rate in HeLa cells (*Gilbert, et al., 2005*; *Moran, et al., 1996*; *Symer, et al., 2002*). HeLa cells differ physiologically in many respects from normal human cells, and outside of HeLa cells, studies of inherited, de novo, and somatic insertions in every other biological context to our knowledge have found only the canonical TSD size distribution, including:

a. Known human-disease causing L1 insertions (19 insertions; TSD mean: 14 bp; range: 2–20 bp, except 4 without a TSD) (*Hancks and Kazazian, 2012*).

b. Tumor somatic insertions (1450 insertions; TSD peak: 15 bp; 98% are $\leq$ 20 bp) (*Helman et al., 2014*; *Tubio et al., 2014*).

c. Comprehensive profiling of population-polymorphic insertions in the 1000 genomes project (TSD peak: 15bp $\pm$ 7 (SD); nearly all between 5 and 25 bp) (*Stewart et al., 2011*).

d. Genome-wide analysis of all L1s in the human genome reference (TSD peak: 15 bp; nearly all < 25bp) (*Kojima, 2010*; *Szak et al., 2002*).

e. Somatic insertions in neurons: six L1 insertions in rat neuronal progenitors in vitro (*Muotri et al., 2005*) and the two bona fide human brain somatic insertions we have found (*Evrony et al., 2012*; *Evrony et al., 2015*) all had TSD sizes < 20 bp.

## Appendix 2: Single-cell genome quality and amplification performance in Upton, et al.

How widely and evenly the genome is amplified (breadth and uniformity) are two measures of single-cell genome quality and amplification performance that are integral to the feasibility and success of subsequent somatic mutation analyses (*Leung et al., 2015*). Each can be assessed at different genomic scales (i.e. resolutions) or sequence elements, for example, in large 500 kb bins, at base-pair resolution, stratified by GC-content, at retrotransposon elements, etc. (*Evrony et al., 2015*; *Zhang et al., 2015*). Breadth and uniformity of genome amplification are related and can vary significantly depending on the genomic scale or sequence element evaluated. For instance, if a hypothetical single-cell genome amplification method amplifies only one small 1 kb region every 500 kb across the genome while the remaining genome is unamplified (i.e. extreme non-uniformity), then a low-resolution analysis of breadth of amplification will show 100% genome coverage at 500 kb bins even though at base-pair resolution only 0.2% (1/500) of the genome is amplified. Such a single-cell method would be suitable for analysis of copy number variants larger than 500 kb, but would not be suitable for smaller mutations. Single-cell amplification breadth and uniformity of coverage must therefore be evaluated at genomic size scales and sequence elements relevant to the type(s) of mutation(s) being studied. We profiled the MALBAC-amplified single neurons from Upton, et al. using previously developed single-cell quality metrics (*Cai et al., 2014*; *Evrony et al., 2015*; *Zhang et al., 2015*), and compared them to MDA-amplified single neurons and MALBAC-amplified single cells from prior studies.

## I. Single-cell quality control performed by Upton, et al.

Upton, et al. initially evaluated single-cell quality and MALBAC amplification by low-coverage sequencing at a resolution of 600 kb bins, estimating 8% allelic dropout and 0.8% locus dropout. However, a scale of 600 kb bins is larger than the size of retrotransposon insertions (< 6 kb) or the DNA fragments used to detect them (< 300 bp). Upton, et al. then evaluated single-cell quality at higher resolution by detection of reference L1-Ta insertions, a genomic scale that is relevant to retrotransposon analysis. Single-cell 5' junction PCR of a set of germline heterozygous L1 insertions detected 50% of junctions. Therefore, Upton, et al.'s MALBAC single cells have greater dropout of L1 element junctions (50%) than the dropout measured at 600 kb bins (0.8%), indicating non-uniformity of genome coverage at small genome scales relative to large genomic scales. This illustrates the importance of analyzing genome amplification uniformity. In contrast, prior MDA single-neuron studies analyzed amplification uniformity and yielded consistent estimates of 8–10% allelic and ~2% locus dropout across the full range of genomic scales, including 500 kb bins, microsatellites (<400 bp), single nucleotide variants, genome-wide base-pair resolution, and direct measurement of L1 dropout by PCR (*Evrony et al., 2012*; *2015*).

## II. Genome coverage

We analyzed breadth and uniformity of genome amplification of Upton, et al. single cells at high and low resolutions (small and large genomic scales), and compared them to MALBAC single cells from *Zong et al. (2012)* and MDA single neurons from *Evrony et al. (2012)*, *(2015)*.

## High-resolution genome coverage

Visualization of genome coverage at base-pair resolution reveals striking non-uniformity in pooled hippocampus and pooled cortex MALBAC single neurons from Upton, et al., with ~1 kb peaks of high coverage separated by troughs of low coverage/dropout (*Figure 4A*). The similarity of the peaks and troughs in hippocampus and cortex single neurons indicates systematic non-uniformity. Remarkably, MALBAC single cells from Zong, et al. have peaks and troughs of coverage in similar locations (*Figure 4A*), indicating the non-uniformity is due to MALBAC amplification. In contrast, at this resolution MDA single-cell samples show significantly better uniformity (*Figure 4A*). The non-uniformity of MALBAC amplification at small genomic scales (< 30 kb) encompassing the size of L1 insertions has been identified by prior studies (*Evrony et al., 2015*; *Zhang et al., 2015*) and likely explains several performance metrics of single-cell RC-seq: a) the low success rate (50%) of PCR for germline L1s; b) the subset of germline gold-standard known non-reference (KNR) insertions appearing at low read counts (*Figure 2A*, label 'a'); and c) the low sensitivity (high dropout) for germline KNR insertions (*Figure 3A* and *Figure 3—figure supplement 1*).

In terms of breadth of coverage at base-pair resolution (the highest possible resolution), Upton, et al. pooled MALBAC hippocampus single neurons capture 89% and 64% of the genome at $\geq$1x and $\geq$10x read depth, respectively, at a total genome-wide average sequencing depth of 30x (*Figure 4—figure supplement 2A*). At the same 30x genome-wide average sequencing depth, MDA single neurons capture more of the genome, with 97% and 73% of the genome at $\geq$1x and $\geq$10x read depth, respectively (*Figure 4—figure supplement 2A*). The pooled MALBAC single-neuron sample from Upton, et al. has similar breadth of coverage as individual MALBAC single cells from Zong, et al. Notably, MDA single neurons achieve greater breadth of genome coverage than MALBAC single cells from both studies across all increasing subsampled read depths (*Figure 4—figure supplement 2A*), indicating consistently lower locus dropout for MDA relative to MALBAC. The plateau of the $\geq$1x genome coverage curves with increasing subsampled read depths estimates a locus dropout of ~3% in MDA single neurons, ~7% in Zong, et al. MALBAC single cells, and ~9% in Upton, et al. pooled MALBAC single neurons. These comparisons indicate similar performance of MALBAC amplification in the two studies but do not mean that individual single cells in the two studies achieved similar genome coverage, because the breadth of genome coverage for pooled Upton, et al. single neurons is likely an overestimate of coverage in individual single cells due to pooling canceling stochastic noise/dropout of individual cells.

## Low-resolution genome coverage

We evaluated uniformity of genome coverage/amplification in individual Upton, et al. single neurons at larger genomic scales of ~500 kb bins (low-resolution view) using median absolute pairwise deviation (MAPD) and median absolute deviation from the median (MDAD) scores (*Cai et al., 2014*; *Evrony et al., 2015*). Lower scores indicate more uniform coverage. This analysis was performed for individual single hippocampal neurons without pooling, since the low-coverage WGS is sufficient for low-resolution analysis of individual cells. Upton, et al. MALBAC-amplified single neurons have both greater variability among cells and significantly lower average quality of genome coverage/uniformity (MAPD $0.53 \pm 0.16$) than both MALBAC-amplified single cells from Zong, et al. (MAPD $0.18 \pm 0.06$) and MDA-amplified single neurons (MAPD $0.33 \pm 0.06$) (*Figure 4—figure supplement 1A*). The lower quality of Upton, et al. MALBAC single cells was unexpected given prior analyses showing that MALBAC single cells (from Zong, et al.) have better coverage uniformity at large genomic scales than MDA (*Figure 4B* and *Figure 4—figure supplement 1A–B*) (*Evrony et al., 2015*; *Zhang et al., 2015*). MDAD scores show the same findings, and these results are independent of total sequencing depth, which minimally affects low-resolution analyses due to single-cell amplification noise significantly outweighing Poisson sampling error as supported by subsampling analyses (*Figure 4B* and *Figure 4—figure supplement 1A–B*) (*Evrony et al., 2015*). Interestingly, a sample created by pooling all Upton, et al. hippocampus single neurons shows significantly better uniformity (MAPD 0.12) than the average uniformity of the individual single neurons as well as individual MALBAC single cells from Zong, et al. (*Figure 4B* and *Figure 4—figure supplement 1A–B*). Therefore, the low quality of Upton, et al. single neurons is not due to systematic noise but rather due to stochastic non-uniformity present in each individual neuron, likely preceding MALBAC amplification, that cancels after pooling. This suggests low quality of single cells prior to MALBAC amplification.

## Power spectral density and Lorenz curve analysis of genome coverage

The above analyses evaluated single-cell genome coverage uniformity at two specific genomic scales/resolutions– single base pairs (high resolution) and 500 kb bins (low resolution). We evaluated uniformity of pooled Upton, et al. single neurons from hippocampus and cortex across the full range of genomic scales using two methods introduced by *Zong et al. (2012)*: power spectral density analysis and Lorenz curves. We previously performed these analyses to compare MDA and MALBAC performance (*Evrony et al., 2015*).

Power spectral density plots show the degree to which variability in read depth (uniformity) is distributed across genomic scales (frequencies). At larger genomic scales (smaller frequencies), MDA single neurons have lower uniformity than MALBAC single cells from Upton, et al. and Zong, et al. At smaller genomic scales the situation is reversed: MALBAC samples from both Upton, et al. and Zong, et al. show a similar peak of increased non-uniformity below a frequency of $~3.5 \cdot 10^{-5}$ bp (i.e. < ~30 kb) (*Figure 4C*), while MDA samples have higher uniformity similar to bulk samples (*Figure 4C*). The power spectral analysis therefore recapitulates the findings of the analyses performed above at specific genomic scales while providing a more comprehensive view of uniformity across genomic scales.

Lorenz curves provide another view of genome coverage uniformity by plotting the cumulative fraction of reads as a function of the cumulative fraction of the genome covered at increasing read depths. Perfectly uniform coverage across the genome would approximate the y = x line. Upton et al. pooled hippocampal neurons show lower overall uniformity than MDA single neurons but better uniformity than MALBAC cells from Zong et al. (*Figure 4B*). The latter is likely due to pooling of Upton et al. single neurons removing their stochastic noise whereas Lorenz curves of individual MALBAC cells from Zong et al. also reflect stochastic single cell noise.

## IV. Conclusions

Our analyses of Upton, et al. single cells confirm the better uniformity of MDA at small genomic scales (<50 kb) suitable for study of small mutations, for example SNVs and retrotransposons, whereas MALBAC has better uniformity at larger genomic scales suitable for study of large copy number variants (*Evrony et al., 2015*; *Hou et al., 2013*; *Zhang et al., 2015*). DOP-PCR methods not presented here have better uniformity than either method at large genomic scales but lower uniformity at smaller genomic scales, making it suitable mostly for large (>50 kb) copy number variants (*Navin and Hicks, 2011*; *Navin et al., 2011*). Because each single-cell amplification method has advantages in different domains, the choice of method depends on the type of mutation being studied– there is no current method ideally suited for all types of mutations. Rigorous evaluation of dropout and uniformity at the size scale of the somatic mutation being studied is imperative prior to undertaking costly downstream single-cell sequencing and somatic mutation analyses.

Our low-coverage analyses of Upton, et al. single neurons also revealed lower quality and greater variability among individual cells compared to prior single-cell studies. Quantitative quality control metrics of single cells prior to costly higher coverage or targeted sequencing can reveal differences in quality between tissues and helps exclude low quality tissues and single cells that would otherwise produce high false-positive or false-negative rates.

Note that at small genomic scales, MALBAC single cells from Zong, et al. have significantly better reproducibility, though not uniformity, than MDA (*Evrony et al., 2015*); the loci at which peaks of coverage are found and the shapes of the peaks are highly reproducible between single cells. This suggests one potential advantage for MALBAC over MDA amplification for studying small-scale mutations such as single nucleotide variants if one were interested only in the areas of the genome reproducibly amplified to high coverage by MALBAC (i.e. at the loci with peaks of coverage). In applications seeking to ensure capture and genotyping of the same loci across all profiled single cells, MALBAC's better reproducibility between samples may be advantageous relative to MDA despite MALBAC's lower breadth of coverage and greater non-uniformity at small genomic scales.

## Appendix 3: Single-neuron sequencing does not preferentially amplify chimeras

Upton, et al. state that in single-neuron RC-seq, as opposed to bulk RC-seq, chimeras are preferentially amplified to higher read counts than true insertions. They further assert the same is true generally of single-neuron sequencing, including prior studies (*Evrony et al., 2012*; *Evrony et al., 2015*). However, analysis of Upton, et al.'s single-cell whole-genome sequencing (WGS) data shows directly that the authors' single-cell MALBAC amplification and WGS library preparation methods do not preferentially create and amplify chimeras relative to true genomic fragments. Chimera rates are reflected in the fraction of discordant read pairs (DNA fragments whose sequencing reads from each end of the fragment align distantly in the genome or in wrong orientation to each other) and clipped reads (partially aligning reads, which includes reads with chimera breakpoints occurring within a read rather than between read pairs of a DNA fragment). Discordant and clipped reads in Upton, et al. single cells comprise on average only $0.9 \pm 0.5\%$ (SD) and $4.2 \pm 1.7\%$ of all reads, respectively. Similar fractions of discordant and clipped reads, $2.3 \pm 1.4\%$ and $1.4 \pm 0.3\%$, respectively, were seen in MALBAC single cells from a prior study (*Zong et al., 2012*) (analysis performed in *Evrony et al. (2015)*). If it were true, as Upton, et al. assert, that chimeras amplify to higher levels than true genomic fragments, then these rates would have been significantly greater. The low discordant and clipped read rates in Upton, et al. single cells confirm that chimeras are a small minority of their samples' DNA fragments, and that their MALBAC and WGS library preparation methods do not preferentially amplify chimeras more than true genomic fragments to any significant degree.

Extensive quantitative analyses of chimeras in prior single-cell studies show that chimeras occur at significantly lower rates than non-chimeric fragments also in the MDA method more commonly used for single-cell amplification (*Evrony et al., 2012*; *Evrony et al., 2015*). Only 0.4% of DNA fragments in MDA-amplified single cells are MDA chimeras, reflecting an MDA chimera rate of ~1.2 per 100 kb of amplified DNA (*Evrony et al., 2015*). An additional 2-3% of fragments in single-neuron WGS samples are chimeras created during sequencing library preparation, reflected in $1.9 \pm 0.5\%$ and $0.8 \pm 0.2\%$, discordant and clipped reads, respectively (*Evrony et al., 2015*), which could be mitigated by alternative library preparation methods (*Quail et al., 2008*). Single-neuron WGS (*Evrony et al., 2015*) has no additional steps in which chimeras could form other than MDA and WGS library preparation. This study further presented MDA chimera spatial distributions that elucidate their mechanisms of formation and aid computational filtering. The above analyses and others (*Lasken and Stockwell, 2007*) have therefore shown unequivocally that chimeras are a minority of reads in single-cell sequencing methods generally (i.e. the number of bases amplified between chimera events is on average much larger than the size of sequenced DNA fragments).

Additionally, both Upton, et al. and prior single-neuron sequencing studies (*Evrony et al., 2012*; *Evrony et al., 2015*) show unequivocally that true L1 insertions appear at higher read counts than chimera calls. In prior studies, the score distributions (reflecting read counts) of gold-standard KNR insertions in single cells have a unimodal distribution at high scores (i.e. at high read counts), clearly separable from the remaining calls concentrated at lower scores (reflecting chimeras) (*Evrony et al., 2012*, Figure S4B; *Evrony et al., 2015*, *Figure 2B*). The KNR read count distributions of both Upton, et al. and these prior studies show the same (*Figure 2A*; *Figure 2—figure supplement 1A–C*), contradicting Upton, et al.'s suggestion that chimeras appear at higher read counts than true insertions. Therefore, in these methods and in single-cell sequencing generally, insertion calls supported by single reads would have negligible probability of being true insertions.

Upton, et al. further support their claim that single-neuron sequencing preferentially amplifies chimeras by remarking on the low PCR validation rate among high-scoring final L1-IP candidates of *Evrony et al. (2012)* (low specificity) as an indication that many true candidates

at low scores are being missed (low sensitivity). This confuses specificity with sensitivity: low specificity among candidate mutations does not imply low sensitivity for true mutations. L1-IP's sensitivity is high, in fact significantly higher than RC-seq, as measured using the gold-standard KNR insertion reference (*Figure 3A*; *Figure 3—figure supplement 1*) as well as single-copy (heterozygous and hemizygous) KNR and human genome reference insertions, the most stringent performance measure (*Evrony et al., 2012*; *Evrony et al., 2015*). L1-IP's specificity performance is due to the very small fraction of chimeras that are at higher scores outnumbering the very small number of true somatic insertions, highlighting the challenge of chimeras in single-cell retrotransposon analysis in the setting of low retrotransposition rates. Notably, the challenge of computational specificity was readily overcome with a definitive 3' PCR validation approach with near-perfect sensitivity and high specificity, followed by full-length validation with perfect specificity (*Evrony et al., 2012*). We emphasize that as in any signal-noise problem where true events are distributed at higher signal than false events, lowering a threshold to include more true events at lower scores would only decrease specificity by including a relatively greater number of false-positive calls– i.e. the proportion of chimeras among candidates called with a lower threshold can only be greater than the proportion among candidates called with a higher threshold. Additional PCR validation illustrates this: 24 randomly selected L1-IP candidates supported by one read and 24 candidates randomly selected without any read count filter all failed PCR validation (*Supplementary file 1*, sheet "L1-IP | low-read-count"), illustrating the need for signal modeling based on read counts and other parameters using a gold-standard true mutation reference.

A subsequent single-neuron WGS method was motivated precisely to increase computational specificity by its ability to detect most insertions at both 5' and 3' junctions (*Evrony et al., 2015*). In fact, the four highest scoring candidates found by this method were bona fide somatic insertions, i.e. greatly improved specificity while retaining high sensitivity. For this reason single-cell whole-genome sequencing coupled with full-length validation is the definitive, (albeit a costly) method for the study of somatic retrotransposition. It is also able to profile multiple retrotransposon types (Alu, L1, SVA) as well as other types of somatic mutation (structural variants, single-nucleotide variants) in one experiment. Targeted methods such as L1-IP and RC-seq if used with an appropriate single-cell amplification method such as MDA, careful signal modeling based on true insertions, and rigorous PCR validation, are cost-effective alternatives for high-throughput studies focused on one type of retrotransposon across many cells.

## Appendix 4: L1 qPCR copy number assay

Upton et al. (2015) employ a quantitative PCR (qPCR) method to measure somatic L1 copy number in in the brain. Their qPCR results show relative differences of ~10-30% in somatic L1 copy number among hippocampi of four normal individuals and an individual with Aicardi-Goutieres syndrome (*Upton et al., 2015*; *Figures 4D–E*). This qPCR method has also been used in prior studies, with results showing variable increases (between 5% and up to 45%) in somatic L1 copy number in brains of normal human individuals, in particular the hippocampus, as well as brains of schizophrenia patients (*Baillie et al., 2011*; *Bundo et al., 2014*; *Coufal et al., 2009*). However, there are known limitations of the qPCR method discussed by its originators (*Erwin et al., 2014*; *Reilly et al., 2013*), namely its use of exogenous LINE-1 DNA to calibrate L1 copy number quantification. Furthermore, as discussed below, the qPCR assay may lack specificity for active L1 elements, which would impact its findings.

Below is an alignment of the qPCR assay (ORF2 #1) primers and TaqMan probe to the L1Hs and L1Pa2-5 consensus sequences from RepeatMasker (asterisks denote bases identical across all sequences) (*Appendix 4—figure 1*). Note that only the 3rd base from the 5' end of the qPCR forward primer distinguishes L1Hs elements from older L1Pa elements (bold blue). The reverse primer and probe do not distinguish between the different L1 subfamilies' consensus sequences.

```
ORF2 #1-Forward 5'-TGCGGAGAAATAGGAACACTTTT-3'
ORF2 #1-Probe                                        CTGTAAACTAGTTCAACCATT
L1Hs              TGCGGAGAAATAGGAACACTTTTACACTGTTGGTGGGACTGTAAACTAGTTCAACCATTG
L1Pa2             TGTGGAGAAATAGGAACACTTTTACACTGTTGGTGGGACTGTAAACTAGTTCAACCATTG
L1Pa3             TGTGGAGAAATAGGAACACTTTTACACTGTTGGTGGGACTGTAAACTAGTTCAACCATTG
L1Pa4             TGTGGAGAAATAGGAACACTTTTACACTGTTGGTGGGACTGTAAACTAGTTCAACCATTG
L1Pa5             TGTGGAGAAATAGGAACACTTTTACACTGTTGGTGGGACTGTAAACTAGTTCAACCATTG
                  ** *********************************************************

ORF2 #1-Reverse    3'-AGTCAGTGTGGCGATTCCTCA-5'
L1Hs              TGGAAGTCAGTGTGGCGATTCCTCA
L1Pa2             TGGAAGTCAGTGTGGCGATTCCTCA
L1Pa3             TGGAAGTCAGTGTGGCGATTCCTCA
L1Pa4             TGGAAGTCAGTGTGGCGATTCCTCA
L1Pa5             TGGAAGWCAGTGTGGCGATTCCTCA
                  ****** *****************
```

**Appendix 4—figure 1.** Alignment of the qPCR assay primers and probe to the L1Hs and L1Pa2-5 consensus sequences.

PCR specificity depends significantly more on the 3' ends of primers than their 5' ends, because DNA polymerase can begin replication of the template even when only a 3' portion of the primer is hybridized (*Bru et al., 2008*; *Stadhouders et al., 2010*). In other words, DNA polymerase depends on the 5' end of the primer only to the extent that it aids in hybridization and stability of the 3' end. In this case, the 3'-most 20 bp of the forward primer match equally well to the L1Hs and L1Pa consensus sequences. Therefore, in order for the PCR reaction to specifically and preferentially amplify L1Hs and not L1Pa elements, the stringency of the PCR reaction would need to be calibrated to allow amplification only when at least the 3'-most *21bp* match to the template while not allowing amplification when only the 3'-most *20bp* match to the template (i.e. a single base mismatch stringency for the 3rd base from the 5' end of the 23 bp forward primer). Such a calibration was not performed by Upton et al. and achieving such a precise PCR stringency/specificity is unlikely using this primer design (*Bru et al., 2008*). A prior study recognized the need for L1Hs specificity in this assay and sequenced amplicons from the qPCR reaction, finding that most amplicons corresponded to L1Hs rather than L1Pa (*Coufal et al., 2009*; table S3). However, this result may be because the primers are incorporated into the amplicons during PCR such that the diagnostic base discriminating between L1Hs and L1Pa would be lost and replaced by the primer sequence.

Therefore, the absolute number of non-L1Hs elements amplified by the assay depends on whether the PCR reaction can discriminate templates matching the 3rd base of the forward primer (i.e. matching the L1Hs-specific nucleotide). If the PCR reaction only amplifies templates matching this diagnostic base, 214 L1 elements are predicted to be amplified per the UCSC In-Silico PCR tool, most of which (88%) are L1Hs and the remainder L1Pa2-4. However, without the specificity provided by the 3rd base of the forward primer, 5,166 L1 elements are predicted to be amplified from the human genome, i.e. ~24 times as many L1 elements, of which only 527 (10%) are L1Hs, and the remainder are 1136 (22%) L1Pa2, 2047 (40%) L1Pa3, 1203 (23%) L1Pa4, 178 (3%) L1Pa5, and other older L1 elements. Moreover, the ORF2 #1 Taqman probe matches the majority (71%) of these elements with no mismatches and nearly all (95%) with 1 mismatch. As a result, most of the L1 elements amplified and quantified by the qPCR assay may be older inactive L1Pa elements rather than retrotranspositionally active L1Hs elements.

Because qPCR measures relative changes in copy number, targeting of mostly inactive L1Pa elements would in turn cause a decrease in sensitivity for detection of L1Hs copy number changes. Additionally, any purported *relative* increase in copy number indicated by the assay would mean more dramatic *absolute* increases in L1 copy number, because relative copy number increases are measured relative to the *total* number of L1 elements amplified by the assay. For example, without specificity for L1Hs elements, even a modest 5% relative copy number increase in this assay would correspond to ~258 (5% * 5,166) additional L1Pa/L1Hs insertions per cell. And the 30-40% relative copy number differences shown by the assay in prior studies (*Baillie et al., 2011*; *Upton et al., 2015*) would reflect differences of ~1550–2070 L1Pa/L1Hs insertions per cell. qPCR estimates of hundreds to thousands of additional L1Pa/L1Hs insertions per cell are biologically unlikely due to the immense mutational burden implied and are inconsistent with even the highest prior estimates of somatic retrotransposition. Furthermore, bulk tissue is profiled by this assay, so the above rate would be revised even further upwards for neurons if retrotransposition were occurring at higher rates in neurons than other cell types.

In light of the above, the qPCR and RC-seq results illustrated in *Figures 4D–E* of Upton, et al. (2015) are inconsistent. Comparing individuals CTRL-36 and CTRL-55 in *Figure 4E* shows ~45 and ~15 somatic insertions, respectively, per individual by RC-seq, while the L1 qPCR assay shows a difference of ~30% in L1 copy number between the individuals. However, a 30% *relative* difference in L1 copy number measured by the qPCR assay would be equivalent to an implausible *absolute* difference at least one order of magnitude larger than that measured by RC-seq; i.e. >300 and perhaps up to ~1500 somatic insertions depending on the number of L1 elements amplified by qPCR.

There are inevitable random and frequent systematic sources of error inherent to qPCR, including sample preparation, pipetting, and instrumentation (*Fernandez-Jimenez et al., 2011*; *Kitchen et al., 2010*; *Weaver et al., 2010*; *Whale et al., 2012*). Robust detection of small (< 10%) relative copy number differences is challenging (*Weaver et al., 2010*). Even with sufficient replicates to eliminate random error, systematic error sources may remain. One systematic error may be differential extraction of normalizer genes' DNA (alpha-satellite DNA) relative to L1 DNA from tissue samples; similar biases have been observed in a mitochondrial DNA qPCR assay (*Guo et al., 2009*).

Finally, the large L1 copy number differences suggested by Upton et al.'s qPCR data are unlikely to be explained by non-retrotransposition mediated processes such as structural variation or aneuploidy, as this is inconsistent with recent studies showing that large genome-wide copy number variation and aneuploidy are infrequent enough to account for such large L1 copy number changes, at least in cortical neurons (*Cai et al., 2014*; *Evrony et al., 2012*; *McConnell et al., 2013*). Even if an improved L1 qPCR assay addressing the above issues were designed, it would still be unable to differentiate changes in L1 copy number due to retrotransposition from those not due to retrotransposition (*Reilly et al., 2013*).

