## [Decision Letter]

Thank you for submitting your work entitled "Single-Neuron Genomics: Resolving Rates of Mutation in the Brain" for consideration by *eLife*. Your article has been reviewed by three peer reviewers, and the evaluation has been overseen by Sean Eddy (Reviewing Editor) and a Senior Editor.

The following individuals involved in review of your submission have agreed to reveal their identity: Jay Shendure, Guillaume Bourque, and Ira Hall (peer reviewers).

The reviewers have discussed the reviews with one another and the Reviewing Editor has drafted this decision to help you prepare a revised submission.

Summary:

The importance of somatic L1 transposition in neurons is debated. The arguments depend on the estimated rate of transposition events. Technically challenging single neuron genomics experiments have given discrepant estimates of <1 new L1 insertion per neuron (Evrony et al. 2012; 2015) versus 13-16 insertions per neuron (Upton et al., 2015). Here, Evrony et al. present a thorough critique of the data analysis and conclusions of the Upton paper, arguing that the Upton analysis overestimated the rate by >50x, and that the actual rate is about 0.2 insertions per neuron, thus resolving the discrepancy with previous results.

The reviewers and Reviewing Editor unanimously agree that your manuscript makes a convincing case. The burden of proof is high to publish such a direct and strongly-worded refutation of another group's results, but we agree that your manuscript meets that bar, and that this important and careful critique is strongly deserving of publication in *eLife* after minor revision.

Required revisions:

We would like you to address the following two points in a revision:

1) The paper's main point should be made more succinctly. The most convincing section of the analysis is the clear difference between the read count data supporting "known germline insertions" (KNR) and novel insertion candidates. In general, the rest of the manuscript, and particularly the appendices, feels somewhat bloated. Specifically: a) the section toward the end of the manuscript on MALBEC vs. MDA could be significantly shortened and b) the "framework for single-cell genomics" in the final section of the manuscript seems like an attempt to spin a larger story out of a single case of an apparently flawed data analysis. This single case does not necessarily reflect a larger problem in the field that justifies a broad call for standardization, and the call for "shared standards" feels premature, given the rapid changes in the field. The paper would be stronger if this section were deleted or condensed to at most a few sentences.

2) You should soften the tone of the paper so that it's less likely to be perceived as combative. Your points are overwhelmingly and substantively supported. They will stand strongly on their own without rhetoric.

---

## [Author Response]

*Required revisions: We would like you to address the following two points in a revision: 1) The paper's main point should be made more succinctly. The most convincing section of the analysis is the clear difference between the read count data supporting "known germline insertions" (KNR) and novel insertion candidates. In general, the rest of the manuscript, and particularly the appendices, feels somewhat bloated. Specifically: a) the section toward the end of the manuscript on MALBAC vs. MDA could be significantly shortened and b) the "framework for single-cell genomics" in the final section of the manuscript seems like an attempt to spin a larger story out of a single case of an apparently flawed data analysis. This single case does not necessarily reflect a larger problem in the field that justifies a broad call for standardization, and the call for "shared standards" feels premature, given the rapid changes in the field. The paper would be stronger if this section were deleted or condensed to at most a few sentences.*

We made further edits for clarity and length, including shortening the MALBAC vs. MDA section and appendices. We also deleted from the Abstract, Impact statement, summary sentence of the Introduction, and final section of the text any mention of standards for the field. We shortened and modified the final section to provide a few suggestions for the design of single-cell genomics studies rather than standards.

2) You should soften the tone of the paper so that it's less likely to be perceived as combative. Your points are overwhelmingly and substantively supported. They will stand strongly on their own without rhetoric.

We thank you for the suggestion and have tried to remove any phrases that may seem harsh or extraneous to the science and data.